

# Probabilistic landslide ensemble prediction systems: Lessons to be learned from hydrology

Ekrem Canli[1], Martin Mergili[1,2], Thomas Glade[1]

**[1]** Department of Geography and Regional Research, University of Vienna, Universitätsstr. 7, 1010 Vienna, Austria

**[2]** Institute of Applied Geology, University of Natural Resources and Life Sciences, Gregor-Mendel-Straße 33, 1180 Vienna, Austria

*Correspondence to*: Ekrem Canli (ekrem.canli@univie.ac.at)

**Abstract.** Landslide early warning has a long tradition in landslide research. Early warning can be defined as the provision of
timely and effective information that allows individuals exposed to a hazard to take action to avoid or reduce their risk and
prepare for effective response. In the last decade, hydrological forecasting started operational mode of so called ensemble
prediction systems (EPS) following on the success of the use of ensembles for weather forecasting. Those probabilistic
approaches acknowledge the presence of unavoidable variability and uncertainty at larger scales and explicitly introduce them
into the model results. Now that convective-scale numerical weather predictions and high-performance computing are getting
more common, landslide early warning should attempt to learn from past experiences made in the hydrological forecasting
community. This paper reviews and summarizes concepts of ensemble prediction in hydrology and how ties to landslide
research could improve landslide forecasting. Three future research directions were identified: 1.) evaluation of how and to
what degree probabilistic landslide forecasting improves predictive skill; 2.) adaptation and development of methods for
validating and calibrating probabilistic landslide models; 3.) application of data assimilation methods to increase the quality
of physical parametrization and increased forecasting accuracy.

**Keywords:** ensemble prediction systems, probabilistic forecasting, landslide early warning

## 1. Introduction

Landslide prediction at regional scale is a hot topic within the scientific community as the time-varying aspects of landslide
susceptibilities, hazards and even risks are crucial for emergency response planning and protecting public safety (Baum et al.,
2010, Glade and Crozier, 2015). Further, the number of landslides is assumed to increase due to global change (Crozier, 2010,
Gariano et al., 2017, Papathoma-Köhle and Glade, 2013). This calls for an increased demand in early warning procedures with
the aim of issuing timely warnings of an upcoming hazardous event to temporarily reduce the exposure of vulnerable persons
or infrastructure (Thiebes and Glade, 2016). In this paper, we use prediction systems synonymously with early warning systems
for terminological consistency within the landslide community although we acknowledge that early warning should also cover
dissemination and response strategies (UNEP, 2012). Warnings can be considered as calls for the public to take protective
action, and the time scale of a warning depends on the associated weather event (Stensrud et al., 2009). For natural hazard



types with a rapid onset, such as shallow landslides and flash floods, rainfall can be considered as the main triggering mechanism. Rainfall nowcasting, or short-term rainfall forecasting, is an important and well-established tool for numerical weather predictions (NWP) in meteorological and hydrological applications that offer rainfall predictions several hours ahead. While it is evident that processes with potentially very short response time require more effort for timely early warning than

just real-time measurement of rainfall, real forecasting initiatives are scarce especially in the landslide community (Tiranti et al., 2017).

The reasons for the rare application of NWP products within the landslide early warning community are manifold. One reason might be the complexity of single landslide detachments: the same landslide triggering event does not necessarily cause other landslides as the time between propagation stage and the collapse phase may vary significantly based on differences in local

conditions of topography, materials such as soil, regolith and rock, vegetation, etc. and spans from minutes (e.g. flow slides on slopes covered with shallow coarse-grained soils) to years (e.g. earth flows in slopes of fine grained soils) (Greco and Pagano, 2017). Based on empirical-statistical relationships between landslide occurrence and its associated rainfall event, rainfall thresholds within a certain confidence interval aim at accounting for those differences in slope failure behavior (Glade, 2000). Guzzetti et al. (2007) give an overview of rainfall and climate variables used in the literature for the definition of rainfall

thresholds for the initiation of landslides, however, such empirical-statistical approaches only pose a simplification between rainfall occurrence and the physical mechanisms leading to landslides, neglecting local environmental conditions and the role of the hydrological processes occurring along slopes (Reichenbach et al, 1998, Bogaard and Greco, 2017). Attempts to relate landslide-triggering thresholds to weather and other physically based characteristics can be very challenging given the quality of currently available data (Peres et al., 2017). Another reason for the negligence of physically based forecasting initiatives

used to be the lacking spatial resolution and computational power for considering such convective-scale phenomena which are of particular interest for modelling small scale related phenomena with a rapid onset such as shallow landslides and flash floods. This became, however, increasingly less of an issue. Convective-scale NWP with spatial resolutions of 1 to 4 km issued in very short time intervals are already available in many parts of the world. The hydrological community has recently adopted to those advancements by implementing such convective-permitting models into operational flood prediction systems

(Hapuarachchi et al., 2011, Liu et al., 2012, Yu et al., 2015).

This paper reviews and summarizes concepts of ensemble prediction systems (EPS) in hydrology and how those can be translated to be applicable also in process-based landslide early warning systems. A strong emphasis is put on how to deal with spatial uncertainties by demonstrating the benefits of probabilistic model application which does not eliminate uncertainty, but it explicitly introduces in into the model results. In a case study, we highlight possible spatially distributed physically based

landslide early warning products for decision makers and point out specific challenges that landslide research has to face in the upcoming years. The aims of this paper are:

    a)   to critically evaluate the current state of physically based landslide early warning, its limitations and possible ties to hydrological forecasting;



b) on this basis, to foster cooperation across disciplinary boundaries to bring together scientists from different fields to pursue research based on forecasting experiences gained in the last couple of years.

## 2. Probabilistic forecasting in hydrology and ties to landslide research

When considering ensemble prediction systems (EPS), one should clarify what is expressed with the term *ensemble* and why EPS should be used at all since it is virtually unused in the landslide community. In the *Guidelines on Ensemble Prediction Systems and Forecasting* issued by the World Meteorological Organization (WMO, 2012), EPS are defined as "numerical weather prediction (NWP) systems that allow us to estimate the uncertainty in a weather forecast as well as the most likely outcome. Instead of running the NWP model once (a deterministic forecast), the model is run many times from very slightly different initial conditions. Often the model physics is also slightly perturbed, and some ensembles use more than one model within the ensemble (multi-model EPS) or the same model but with different combinations of physical parametrization schemes (multi-physics EPS). […] The range of different solutions in the forecast allows us to assess the uncertainty in the forecast, and how confident we should be in a deterministic forecast. […] The EPS is designed to sample the probability distribution function (pdf) of the forecast, and is often used to produce probability forecasts – to assess the probability that certain outcomes will occur" (WMO, 2012, p. 1).

Krzysztofowicz (2001) argues that forecasts should be stated in probabilistic, rather than deterministic, terms and that this "has been argued from common sense and decision-theoretic perspectives for almost a century" (Krzysztofowicz, 2001, p. 2). But still, by the new millennium, most operational hydrological forecasting systems relied on deterministic forecasts and there was a too strong emphasis on finding the *best* estimates rather than quantifying the predictive uncertainty (Krzysztofowicz, 2001). However, those times have been overcome a decade later (Cloke and Pappenberger, 2009). From a scientific and historical perspective, landslide prediction has very strong roots in empirical-statistical threshold based approaches (Wieczorek and Glade, 2005, Guzzetti et al., 2007). This stands valid until today, since most operational landslide early warning systems rely purely on the relationship between rainfall and landslide occurrence, thus representing only a simplification of the underlying physical processes. Baum and Godt (2010), Alfieri et al. (2012a), Thiebes (2012) and Thiebes and Glade (2016) give an overview of present and past operational landslide early warning systems (EWS). Bogaard and Greco (2017) critically analyze the role of rainfall thresholds for shallow landslides and debris flows from a hydro-meteorological point of view.

One reason why landslide forecasting is seemingly more challenging can be attributed to the spatial and temporal predictability of landslide processes. The spatial occurrence of floods is topographically foreseeable and controllable which is much more difficult to assess for landslides in distributed modelling due to their very localized nature (Alfieri et al., 2012a, Canli et al., 2017). Also, the prediction domain in flooding, which is usually streamflow, is rather straightforward to observe and to be measured accurately over a long time. In the past 15 years, a mindset of adapting probabilistic concepts to account for inherent uncertainties has taken over in the hydrologic community and the move towards ensemble prediction systems (EPS) in flood





forecasting represents the state of the art in forecasting science, following on the success of the use of ensembles for weather forecasting (Buzzia et al., 2005, Cloke and Pappenberger, 2009).

Unfortunately, initiatives such as the *Hydrological Ensemble Prediction Experiment* (HEPEX) were not fostered in the landslide community to date. The general aims of this ongoing bottom-up initiative are to investigate how to produce, communicate and use hydrologic ensemble forecasts in a multidisciplinary approach (Schaake et al., 2007). One reason for the absence of such cooperative efforts might be the political, and therefore also financial, situation that led to the advancement of ensemble predictions in hydrology. Many international bodies demonstrated their interest in EPS which led to this superior position of hydrological prediction. This is even more so the case when taking into account transboundary floods that are typically more severe in their magnitude, affect larger areas and cause more damage and overall losses (Thielen et al., 2009). Beven (1996) argues that the importance of water resources management led to considerably higher efforts by both researchers and government agencies in hydrological data collection.

Losses from landslides are perceived as mainly private and localized economic losses and thus, only few public resources have been allocated to develop sound spatial landslide early warning systems (Baum and Godt, 2010). As a result, spatial operational landslide early warning systems are scarce and many of them never surmounted their prototype status. Consequently, long monitoring time series, which are indispensable for sound and reliable early warning systems (such as available e.g. for floods, storms, etc.), are commonly not available. Additionally, methodological issues or inadequate monitoring together with insufficient warning criteria significantly reduce the ability of existing systems to issue effective warnings (Baum and Godt, 2010). When looking at the raw numbers, hydrological events rank among the main disaster events together with meteorological events when comparing events in global and multi-peril loss databases, while geophysical events take only a small fraction in absolute numbers (Alfieri et al., 2012a, Wirtz et al., 2014). However, it is widely accepted that landslide losses are vastly underestimated (Petley, 2012). There are several reasons for this observation: a) major disaster databases, e.g. the NatCatSERVICE from the reinsurance company Munich Re, associate landslides as subordinated hazard types of geophysical (amongst earthquakes) or hydrological hazards (amongst floods or avalanches) (Wirtz et al., 2014); b) landslide databases are inconsistent, incomplete or entirely absent and most of the existing inventories severely lack historical data (Wood et al., 2015).

## 3. Benefits and types of probabilistic approaches

Generally speaking, in an ensemble forecast small changes (perturbations) are made to the model parameters and then the model is re-run with these slightly perturbed starting conditions. If the different model realizations (*ensemble members*) are similar to each other, the forecasting confidence is rather high. Contrary, if they all develop differently, the confidence is much lower (WMO, 2012). By considering the proportion of the ensemble members that predict a storm or a landslide, we can make an estimate of how likely the storm or landslide occurs.



The term *ensemble prediction* for environmental applications was coined in the field of meteorology, thus describing the application of numerical weather prediction systems, but it is used in different ways in neighboring disciplines. The atmospheric component is consistently described as weather ensemble input, yet the same applies to how observations of the land surface are incorporated into distributed forecasting models. In the data assimilation stage, ensembles of plausible land

surface state observations (initial streamflow, soil moisture, snowpack, etc.) are created. Using multiple feasible parameter sets for each model or for each model run will realistically increase the spread of possible outcomes, yet it is more objective in terms of considered input parameters that were not directly observed (Schaake et al., 2007). Thus, the term ensemble prediction may be used in any instance of multi-parametric or multi-model data input that is used for forecasting the target variable.

In landslide research, there are a few attempts that explicitly address ensemble techniques as a means of overcoming limitations from purely deterministic approaches or by increasing the predictive performance of statistically based susceptibility mapping. None of them, however, incorporate ensemble techniques in real-time applications. Pradhan et al. (2017) used an ensemble approach to evaluate the output of a physically based model for a statistical machine learning model in varying hydrological conditions. Their ensemble model is based on a maximum entropy model that creates and combines multiple models to improve

modeling results. However, their distributed output does not predict when or exactly where landslide will occur, but yields a classified map with information where landslide occurrence can be expected over the long-term. Thus, their presented ensemble approach indicates landslide susceptibility that may be applicable for regional/spatial planning. While the term *ensemble* is by no means used a lot in landslide studies, it seems that it is predominantly used by the statistical landslide susceptibility modeling community (e.g. Lee and Oh, 2012, Althuwaynee et al., 2014a, Althuwaynee et al., 2014b). It is,

however, not used in any way to address uncertainties in a forecasting model (Bartholmes and Todini, 2005, Vincendon et al., 2011). In a very promising approach, Chen et al. (2016) couple a deterministic model with probabilistically treated geotechnical parameters with rainfall input from an operational multi-scale and multi-member NWP system (GRAPES) to forecast spatial landslide occurrences with their ensemble prediction model (GRAPES-Landslide).

While there are not many landslide studies using or at least addressing ensemble techniques, there has been quite some work

done on probabilistic landslide hazard analysis in the recent past. Lari et al. (2014) propose a probabilistic approach expressing hazard as a function of landslide destructive power where landslide intensity (in terms of displacement rate) is considered rather than their magnitude. Haneberg (2004), Park et al. (2013), Raia et al. (2014), Lee and Park (2016) and Zhang et al. (2016) treat soil properties at regional scale applications in a probabilistic way by randomly selecting variables from a given probability density function, mostly by means of Monte Carlo (MC) simulation. Salciarini et al. (2017) tried to enhance those

approaches by considering geostatistical methods to provide the spatial distribution of soil properties and by using the Point Estimate Method (PEM) as a computationally more efficient method compared to MC simulation. But still, none of those probabilistic approaches are operated in spatial real-time early warning systems, not even on a prototype basis. The research of Schmidt et al. (2008) represents a remarkable exception: they proposed a coupled regional forecasting system in New



Zealand based on multiple process-based models (NWP, soil hydrology, slope stability). Unfortunately, a continuation of this research was not further pursued.

In general, it is possible to distinguish between three types of EPS: global, regional and convective-scale EPS. They each address different spatial and temporal scales in the forecast. For rainfall-induced landslide applications, the latter is the most appealing; thus, we will focus on this one alone. Convective-scale NWP, with model grid sizes of 1–4 km, can attempt to predict details such as the location and intensity of thunderstorms (WMO, 2012). Therefore, those systems reduce the effect of highly intermittent rainfall events that cause serious issues with small-scale rainfall events when applying geostatistical rainfall interpolation techniques (Canli et al., 2017). Convective-scale NWP models are likely to better resolve the intensity and spatial scale of local precipitation, especially in convective precipitation when topographic forcing is involved. Therefore, they are particularly valuable for predicting small scale phenomena, such as flash floods or landslides. However, the major drawback of convective-scale EPS is the immense cost of running (WMO, 2012).

In the past 15 years, many experimental and operational mesoscale EPS have been developed, yet very few with regard to convection-permitting EPS. In, 2012, the German Weather Service (Deutscher Wetterdienst - DWD) started operational mode for their COSMO-DE-EPS with a resolution of 2.8 km (Baldauf et al., 2011, Gebhardt et al., 2011). Similar operational forecasting systems have been implemented in the last couple of years by the weather services of France using their 2.5 km AROME model (Seity et al., 2011), the UK with their 2.2 km MOGREPS-UK model (Golding et al., 2016) and the USA using the 3 km High Resolution Rapid Refresh (HRRR) model (Ikeda et al., 2013).

## 4. The hydrological equivalent of rainfall-induced shallow landslides: the case of flash floods

One major difference between flood and landslide early warning is the available lead time. While the lead time in larger river basins is sufficiently long to prevent any hazardous situations from river flooding, shallow landslides, in the case of first time failures, generally occur suddenly and spatially unforeseeable in a specific area susceptible to landsliding. As opposing to regular floods, however, flash floods can indeed be considered as an appropriate counterpart to rainfall-induced shallow landslide occurrence. Flash floods are, similar to shallow landslides, characterized by the superior importance of small-scale extreme precipitation events and their rapid onset, which leaves only little response time. it is therefore appropriate to examine how flash flood forecasting is performed and how it is applicable to landslide forecasting. What makes landslide forecasting particularly challenging is the evolutionary sequence of the process.

Greco and Pagano (2017) distinguish between three stages of a typical predictive system's architecture: I) the predisposing stage, II) the triggering and propagation stage, and III) the collapse stage. While in hydrological applications (II) and (III) are hardly distinguishable from each other, for rainfall-induced landslides this is not necessarily the case. While the predisposing stage (I) is determined by e.g. increasing pore water pressure due to a varying length of rainfall input that worsens the slope stability conditions, the triggering and propagation stage (II) spans from first local slope failures until the formation of associated slip surfaces. The collapse phase (III) ultimately consists of the mobilization of the entire mass leading to the actual



failure. However, the time between stages (II) and (III) may vary significantly based on differences in local geomorphology, soil, vegetation, etc. and spans from a couple of minutes (e.g. flow slides in slopes covered with shallow coarse-grained soils) to years (e.g. earth flows in slopes of fine grained soils) (Greco and Pagano, 2017). Even when spatially distributed process-based landslide predictions are performed in relatively homogeneous regions, this time offset still prevails and makes landslide

modelling in any context a challenging task. Therefore, warnings should generally be issued during indications of stage (II) since the lead time of stage (III) might be too short given the rapid kinematic characterization of the post-failure behavior, as recent disastrous examples in Italy have shown (Greco and Pagano, 2017).

Hydrological forecasting systems relying only on rainfall observations do not allow for a sufficiently long lead time for warnings. Extending this forecasting lead time further than the watershed response times requires the use of quantitative

precipitation forecasts (QPF) from numerical weather predictions (NWP) (Vincendon et al., 2011). Additionally, models to represent hydrologic and hydraulic processes within a catchment to determine how rainfall-runoff accumulates is required (Hapuarachchi et al., 2011). With regard to producing quantitative precipitation estimates (QPE) in real-time, research has gone into blending multiple sources of information (radar, satellite and gauged data) to increase the accuracy of QPEs. This process is generally referred to as *data assimilation* and is considered as increasingly important for improving hydrological

predictions (Reichle, 2008).

For predicting flash floods, however, longer lead times are necessary and thus high resolution QPFs with 1–6-hour lead times are generated.  In recent years, the spatial (<5 km) and temporal (<1 h) resolutions of NWP model rainfall forecasts have significantly improved, while the combination of such NWP model forecasts with blends of the advected patterns of recent radar, satellite and gauged rainfall data additionally increased the accuracy of nowcasting products (Hapuarachchi et al., 2011).

Based on those high-resolution NWP model forecasts, probabilistic ensemble prediction systems have aided in exploring and quantifying uncertainties. Numerous studies have used those probabilistic precipitation forecasts to drive hydrological models (Vincendon et al., 2011, Bartholmes and Todini, 2005, Siccardi et al., 2005, Thielen et al., 2009). The application of such convective-permitting ensemble NWP is computationally very demanding and still in its infancy with respect to flash flood prediction (Alfieri et al., 2012b). However, a further reduction of the spatial uncertainties of high-resolution rainfall fields is

highly desirable, given the fact that rainfall is still considered as the most uncertain parameter in hydrological forecasting systems (Hapuarachchi et al., 2011, Alfieri et al., 2012b).

## 5.  Many sizes fit all: the concept of equifinality

The concept of equifinality is deeply rooted in the hydrological community. It expresses an acceptance that many sets of parameters may provide equally acceptable forecasts (Beven, 1996, Beven and Freer, 2001, Collier, 2007). The concept of

equifinality revolves around the rejection of the concept of *the* optimal model in favor of multiple possibilities for producing acceptable simulators (Beven and Freer, 2001). This concept is based on the understanding of physical theory and relates to the plethora of interactions among the components of a system whose resulting representations may be equally acceptable.



Research generally follows a working paradigm that should lead to realistic representations of the real processes and characteristics. This idea of identifying a single optimal representation of reality is very distinct in environmental sciences. A major problem arises from the scale discrepancy between sampling and distributed modeling where the use of global parameters undoubtedly leads to errors in predicting local responses at points with unique characteristics (Beven and Freer, 2001). By acknowledging that there are many different model structures or many possible parameter sets scattered throughout the parameter space, the range of predicted variables is likely to be larger than linearized solutions would suggest. This equally means acknowledging that there are uncertainties inherent surrounding the area of parameter space around the *optimum*. As a result, such approaches allow nonlinearity to be considered for predictions (Beven and Freer, 2001).

Geomorphological systems can indeed be considered as transient, inheriting remnants of past and present processes. Environmental systems can exhibit certain degrees of chaotic behavior which results in an inability to express the trajectory of their development based on present-day evidence alone. Therefore, equifinality should not be considered as an indication of a poorly developed methodology, but as something inherent in geomorphological systems (Beven, 1996). However, it should most certainly *not* serve as a loophole for an inadequate methodology or model setup! A practical consequence of this equifinality may lead to a more robust approach to testing the viability of different model setups with the aim to reject some, but to retain many of the offered solutions (Beven, 1996). Similarities and differences in model results should ultimately lead to an improved process understanding and, hence, predictive models with a higher sensitivity and specificity.

## 6. Calibration and validation of probabilistic forecasts

A model is an abstraction and a simplification of reality, hence the need for assessing its validity. Model validation provides a legitimacy in terms of arguments and methods (Oreskes et al., 1994). However, model validation is difficult when the most interesting events are rare, which is generally the case for flash floods or landslides. Also, calibration might be difficult for certain variables, or where suitable observations are not available. The WMO (2012) suggests that direct model output (DMO) from ensembles, although not ideal, still provide valuable information (WMO, 2012). The probabilistic forecasts with a DMO might not be as sharp (e.g. larger ensemble spread), but they still offer an estimate of the uncertainties and thus pose an advantage over purely deterministic forecasts. But even where measurements of modeling parameters are available, it has often shown that those parameters cannot be assumed constant in space or time, which makes calibration even more difficult. Additionally, the scale of measurement generally differs significantly from the scale at which the applied model requires "effective" parameter values to be specified (Beven, 1996).

Deterministic models for landslide prediction synthesize the interaction between hydrology, topography, vegetation and soil mechanics in order to physically understand and predict the location and timing that trigger landslides. These models usually contain a hydraulic and a slope stability component with different degrees of simplification (Formetta et al., 2016). In most cases, the target variable is the slope safety factor (FoS), which is useful as it enables decision makers to take actions when if



falls short of a certain threshold (e.g. FoS < 1.0). Also, when talking about the probability of an event occurring, this event must be defined:

- What is the threshold value to be exceeded?
- What is the exact time or time period to which the forecast refers?
- What is the exact location or area to which the forecast applies?
- Which uncertainties are considered and what is their role in the modelling process?

With regard to those questions and as a starting point, the FoS is a suitable variable for probabilistic forecasting. Yet it has two major flaws: a) it is only a ratio of resisting forces to driving forces that is commonly not directly measured in the field and cannot be directly monitored, and b) landslide events are rare and (unlike streams for example) their future location of occurrence remains unknown until they occur. This makes landslide calibration a really challenging task. And there are limitations of model calibration in the case of rare events. Commonly, calibration will improve the reliability of forecasts (i.e. the match of the target variable or forecast probabilities to frequency of observations of the event) but reduce the resolution of the forecast (the ability to discriminate whether an event will occur or not). Consequently, calibration will improve forecasts of common events, but will also lead to the underprediction of more extreme events. The WMO (2012) argues that this is the case for rare events, since the statistical distributions are trained to the more common occurrences. For rare events, hence, calibration cannot be expected to provide significant improvement over the raw forecasts.

Besides model calibration, validation is an important part within forecasting. Validation unfortunately comes with a rather strong emphasis on either-or-situations. In practice, few (if any) models are entirely confirmed by observational data, and few are completely refuted (Oreskes et al., 1994). On top of that, for most models there may be multiple combinations of parameter values that provide almost equally good fits to the observed data. Thus, changing the calibration period or the goodness-of-fit measure results in an altered ranking of parameter sets to fit the observations. Consequently, there is no single parameter set (or model structure) that serves as the *characteristic* parameter input for any given area, but there is a certain degree of model equifinality involved when reproducing observations with model predictions (Beven, 1996). Therefore, given the issues with multiple (interacting) parameter values, measurement scales, spatial and temporal heterogeneity or the dependence on the model structure, there can never be a single set of parameter values for the calibration process that represents an optimum for the study area, but calibration can contribute to the reduction of range in the possible parameter space.

As a result, this is a field where probabilistic model output really shines, as it expresses the entire model spread with its inherent uncertainties not in absolute terms, but shows the relative performance of a model with respect to observational data. Many decision makers and practitioners in all kind of earth science related fields still favor absolute model output, especially in areas where public policy and public safety is at stake. Unfortunately, certainty is an illusion and ultimately the reason for modeling: the lack of full access, either in time or space, to the phenomena of interest (Oreskes et al., 1994). In practice, there are many measures that attempt to validate probabilistic forecasts. Some are better, some less suitable for distributed model output that is commonly the main form of data representation in landslide early warning. Without going into detail in this paper, we



highlight the work of Mason and Graham (1999) and the WMO (2012) that mention a few skill scores suitable for probabilistic outcomes.

## 7. Case Study

In a simplified ensemble modelling approach applied to a larger study area in Austria (approx. 1366 km²), this specific case study aims to investigate a) how equifinality influences modelling outcome with purely literature based geotechnical parametrization, b) which ways of visual representation are viable for presenting probabilistic data, and c) how infrastructure data can further supplement early warning procedures in an exposure context.

### 7.1 Study Area

The Rhenodanubian Flyschzone (RDF) in the federal state of Lower Austria stretches over approx. 130 km in a SW–NE striking direction. The study area is limited to this geological zone in order to keep the subsurface as homogeneous as possible (Fig. 1). The Cretaceous–early Tertiary RDF is located in the northern foothills of the East Alps, in between the Molasse basin to the North and the Northern Calcareous Alps to the South. The RDF is a paleogeographic-tectonic unit as part of the oceanic Penninic zone that was to a large part eliminated in the subduction process involved in the Alpine orogeny (Hesse, 2011). Flysch materials in the RDF are typically deeply weathered and mainly consist of alterations of pelitic layers (clayey shales, silty shales, marls) and sandstones. Physiographically, the RDF can be characterized as a low mountain region with a highly undulating terrain. It is exceptionally prone to landsliding, exhibiting around five landslides per km2 (Petschko et al., 2014). Heavy rainfall events (exceeding 100 mm per day) as well as rapid snowmelt are considered to be the main triggering factors for slope failure in the region (Schwenk, 1992, Schweigl and Hervás, 2009).

**Figure 1: (A) Location of the Rhenodanubian Flyschzone in Lower Austria (DEM: CC BY 3.0 AT–Federal state of Lower Austria); (B) Typical earth slide in Lower Austria after a heavy rainfall event in May 2014 (Picture: K. Gokesch).**

### 7.2 Modeling Approach

### 7.2.1 TRIGRS

Physically based models used to be attributed to local scale applications (e.g. Corominas et al., 2014, van Westen et al., 2008) because of their computational requirements and data constraints. This has clearly shifted in the last couple of years and by now, physically based models can be quite commonly found to evaluate rainfall-induced landslide susceptibility at the regional scale. The majority is infinite-slope model based with only a few necessary input parameters to be suitable at a regional scale. Increasing the physical basis of a model comes at the cost of introducing even more parameters, while the available data for calibration does not increase at the same time and could lead to problematic overparameterization (Beven, 1996). Even the simplest infinite-slope stability models generally require more parametrization than can be justified by available data.



However, there are some general features of hillslope hydrology that are relevant to slope instability that can be considered to a certain degree by infinite-slope models: vertical infiltration, dependence of infiltration on initial soil moisture conditions, varying time scales for infiltration and lateral flow (Baum et al., 2010). As a result, TRIGRS (*transient rainfall infiltration and grid-based regional slope-stability analysis*; refer to Baum et al. (2008) for details), which we use in this case study, offers a

good trade-off between model complexity and flexibility while we acknowledge the availability of other dynamic, physically based models that were applied at a regional scale, such as STARWARS/PROBSTAB (Kuriakose et al., 2009) or r.slope.stability (Mergili et al., 2014a). Raia et al. (2014) with their TRIGRS_P model and Salciarini et al. (2017) with their PG_TRIGRS model have already attempted a probabilistic TRIGRS derivative in the recent past that gave us the confidence to use TRIGRS in an automated probabilistic approach.

TRIGRS was specifically developed for modeling the potential occurrences of shallow landslides by incorporating transient pressure response to rainfall and downward infiltration processes (Baum et al., 2008). Initial soil conditions are assumed either saturated or tension-saturated. TRIGRS computes transient pore-pressure changes to find analytical solutions to partial differential equations, representing one-dimensional vertical flow in isotropic, homogeneous materials due to rainfall infiltration from rainfall events with durations ranging from hours to a few days. It uses a generalized version of Iverson's

(2000) infiltration model solution to the boundary problem posed by Richard's equation. This solution assesses the effects of transient rainfall on the timing and location of landslides by modeling the pore water pressure of a steady component and a transient component (Liao et al., 2011). However, the model is limited by its distributed one-dimensional modeling approach with noninteracting grid cells and its simplified soil-water characteristic curve (Baum et al., 2010). The entire theoretical basis together with all model related assumptions and equations can be found in Baum et al. (2008, 2010). TRIGRS computes a

factor of safety (FoS) for each grid cell based on an infinite-slope model. It allows for the implementation of spatially varying raster input (e.g. rainfall, property zones, soil depth, infiltration, etc.) to account for horizontal heterogeneity. The FoS can generally be referred to as the ratio of resisting forces (the resisting basal Coulomb friction) over driving forces (the gravitationally induced downslope basal driving stress) on the potential failure surface, with a FoS < 1.0 indicating slope instability and a FoS ≥ 1 slope stability respectively.

**7.2.2 Model Setup**

The probabilistic modeling setup is realized entirely in an open source framework. This was done not only to make it as easily reproducible as possible, but also because it offered the largest flexibility. TRIGRS, which itself is open source, is operated by providing input text files that contain many lines. Those input files are used to specify the numerical values of the input parameters, the location of the input raster files in the filesystem, and all other relevant grids to be considered (e.g. spatially

distributed rainfall maps, different property zones to subdivide the study area in homogenous regions, spatially distributed soil depth maps, etc.). We used python programming language in a script for all string formatting procedures that receives its data from an initialization file. That python script is also used for parsing the raw input into variables usable for TRIGRS. User provided arguments in this initialization file hold the number of property zones needed, the rainfall duration pattern, number





of timesteps and all variables that are used for the probabilistic treatment of parameters, such as min/max values for soil depth, effective cohesion and effective friction angle as well as the number of model runs. The most recent rainfall input can be automatically imported by predefined naming conventions.

We used the GDAL package (GDAL Development Team, 2017) for reading and writing raster files and the NumPy package
(van der Walt et al., 2011) for all raster calculations in the python script. Based on the number of predefined model runs, for each run a single deterministic output is generated based on the selected input parameters derived randomly from a normal distribution. We computed 25 model runs for each hour which resulted in 25 equally probable model results based on the different input parameters. After the initial deterministic model run, a new file is updated after each iteration that is used as the probability of failure (PoF) output. It tracks for each raster cell the initial value of the deterministic factor of safety output, and
in case a cell holds a FoS < 1.0 (unstable cell), the corresponding PoF raster cell receives this information by diving the count of unstable raster cells by the number of model runs in order to calculate a probability value for this raster cell to fail at this location given the different input parameters. All used variables, deterministic model outputs (the FoS maps) and the probabilistic model output (the PoF map) are parsed through to R (R Core Team, 2017). In R, all piped arguments from the python script are used for producing ready-to-use maps (packages: rgdal (Bivand et al., 2017), sp (Pebesma and Bivand, 2005))
or to visualize performance measures such as ROC plots (package: ROCR (Sing et al., 2005)). The entire procedure from importing raw data to producing usable maps is fully automated within an executable file that may be initiated every hour. This open code structure is flexible enough to enable the direct implementation of the most recent available data (rainfall data, soil moisture data, etc.) with minimal effort and thus makes it a useful tool in considering data assimilation techniques.

### 7.2.3 Parametrization

Model parametrization over large areas is a difficult task given the poor spatial comprehension of the spatial organization of involved geotechnical and hydraulic input parameters. Tofani et al. (2017) performed 59 site investigations to parametrize their distributed slope stability model. This amount of in situ soil samplings with associated lab measurements is exceptional and a great source to determine the prescribed probability density function of all measured parameters, especially since all measurements from all sampling sites were published. Although Tofani et al. (2017) ultimately used the median value for each
lithological class, the boxplots suggested normal to lognormal parameter distributions. This is a common observation and might be a result of the central limit theorem, which indicates that lumping data from many different sources (i.e. different in situ soil sampling sites in this case) tends to result in a normal or lognormal distribution (Wang et al., 2015). This gives us confidence to use plausible parameter ranges with a normally distributed state function based on geotechnical textbooks to characterize soils in our study area.

In accordance to the generalized likelihood uncertainty estimation (GLUE) methodology proposed by Beven and Binley (1992), we use a simple Monte Carlo simulation of multiple randomly chosen parameter sets within a predefined parameter range and within a single model structure as the basis for incorporating the inherent parameter uncertainties. Parameters that are considered in a probabilistic way are soil depth, effective cohesion and the effective friction angle (Fig. 2). We assume





fully saturated conditions (θ = 40%) and slope-parallel groundwater flow for the sake of simplicity and given the absence of appropriate initial water conditions. Using all this information, it is now possible to have a spatially distributed probabilistic assessment of the FoS, expressed as the probability of failure (PoF). As TRIGRS is capable of calculating the increase in pore water pressure within the soil, the result is a distributed representation of the decrease in shear strength until slope failure (FoS

< 1.0) is reached at a certain depth.

**Figure 2: Probabilistically derived modeling parameters based on random sampling from a normally distributed state function. Jittering dots (to prevent overplotting) indicate individual samples within a plausible parameter range.**

The raster cell size of the DEM to derive all model relevant topographical parameters used in this case study, is 10 meters.
This cell size allows for a sufficiently high representation surface topography without losing too much information through surface aggregation and smoothing. For the rainfall input, three hourly timesteps were applied with spatially distributed rainfall raster maps representing hourly rainfall based on automated geostatistical interpolation (the methodology is described in detail in Canli et al. (2017)). Using interpolated rainfall input is sufficient as a proof on concept for this case study, but this can be immediately exchanged for any other raster input, such as numerical weather predictions, in a real-time application. The
selection of hourly rainfall input as well as the decision to choose a three-hour timeframe to force the model was made arbitrarily as for the study area there are no published information available on the hydrological response of landslides to rainfall. The spatial resolution of 1 km for the rainfall input was resampled to match the cell size of the DEM, which is a prerequisite of TRIGRS.

## 7.3 Results

Fig. 3 shows the results for 24 model iterations for the same time based on spatially distributed, hourly rainfall input over the last three hours. Each ensemble member was initialized with probabilistically derived parameters that are displayed on each map. The WMO (2012) describes this form of EPS representation *postage stamp map* that shows each individual ensemble member which allows the forecaster to view the scenarios in each member forecast. The results indicate quite significant changes across individual members, but also quite high similarities although parameters change drastically between some of
the members. For example, a depth of 2.5 m, an effective cohesion of 13.4 Nm$^{-2}$ and an effective friction angle of 35 degree in one of the deterministic outputs reveal almost the identical FoS distribution with a depth of 2.0 m, an effective cohesion of 5.4 Nm$^{-2}$ and an effective friction angle of 22.7 degree.

**Figure 3: Postage stamp map for 24 model iterations for the same time. Each ensemble member was initialized with altered
parameters within a plausible range to account for variability and spatial uncertainty. Factor of Safety (FoS) values < 1 indicate slope instability.**

By using a probabilistic representation, this variability and uncertainty is accounted for. Here, the probability is estimated as a proportion of the ensemble members that predict an event to occur (FoS < 1.0) at a specific raster cell. For example, a



probability between 0.75 and 1.0 means that a specific raster cell, under varying input parameters, indicates slope failure in 75% to 100% of all model runs for this specific time.

To provide additional information, which supports different actors responsible to manage landslide hazards, the PoF is underlain with accurately mapped building polygons and roads for a direct exposure visualization of the elements at risk towards landslides (Fig. 4). Buildings and roads are imported from the freely accessible OpenStreetMap (OSM) database. OSM covers almost the entirety of existing buildings in Austria and is based off official Austrian administrative data, which stands under an open government data (OGD) license. Building exposure is a result of a simple spatial join that assigns each building the highest PoF value within 25 m. This value, while arbitrarily chosen, further accounts for spatial uncertainties since TRIGRS models only the location of actual landslide initiation. High building exposure along the river is a modeling artifact introduced by the steep retaining wall and the associated sudden and steep decline in slope angle. Results of the PoF map suggest quite a narrow ensemble spread, which means that the different input parameters indicate an expression of equifinality. This can be considered as some kind of *spatial confidence buffer* that gives some reliance that under varying rainfall forcing the location of possible slope failure is modelled quite consistently at the more or less same location.

**Figure 4: Probability of Failure depicted as a proportion of the ensemble members that predict an event to occur (FoS < 1.0). Building exposure to current slope failure predictions adds an additional information layer for decision makers. Buildings and roads are imported from the freely accessible OpenStreetMap (OSM) database (© OpenStreetMap contributors).**

## 8 Discussion

Since landslides generally tend to occur in steeper slopes (Liao et al., 2011), this spatial confidence buffer modelled in the probabilistic approach presented here could partially alleviate two issues: a) reduce the influence of positionally imprecise landslide inventory data in the calibration process since a larger slope proportion reveals instability; b) reduce the false alarm ratio since landslide locations are more likely to be situated within a certain slope failure probability segment (as would be the case in Liao et al., 2011 for example). In this case study, we can only perform some kind of qualitative validation for the following reasons: a) for Lower Austria a very comprehensive and spatially accurate landslide inventory based on high-resolution airborne LiDAR based DEM mapping exists (Petschko et al., 2015), however, it does not contain any temporal information; b) the Building Ground Registry (BGR) is the most comprehensive source of reported damage causing landslides in Austria, however, its spatial and temporal accuracy is insufficient for physically based model calibration and validation. Qualitative validation by visual comparison (Fig. 5) indicate, for this specific time and under the given rainfall input, that there is an agreement between some of the landslide initiation points and areas of high failure probability.

For personnel responsible to manage landslides in a given region, however, this situation would be quite challenging in order to take appropriate action. The probabilistic approach depicts spatial variability and uncertainty much better than any purely deterministic result, yet there are still many unaccounted uncertainties involved with respect to actual slope failure prediction. Thus, a map representation of slope failure probability at such high spatial resolution could suggest a certainty that simply is



not achievable in landslide modeling. It has to be stressed that this probabilistic approach does not eliminate uncertainty, but it explicitly introduces it into the model results. This is quite detrimental to the ultimate goal of predictive modeling: to be a positionally and temporally accurate mitigation tool. Salciarini et al. (2017) points out that such a tool can be suitable for a first susceptibility screening of an area prone to landsliding, but not for single slope/single landslide analyses. Since such a

map reveals a high degree of spatial discontinuity in its spatial prediction pattern, this undoubtedly puts the forecaster at risk of missing some real landsliding occurrences. This raises the question whether putting high efforts into probabilistic landslide forecasting is warranted compared to a combination of statistical susceptibility maps with an early warning approach including empirical-based rainfall thresholds (see conclusions *Challenge 2: Rare events and model averaging*). Kirschbaum et al. (2012) present such a nowcasting attempt at a regional and global scale by using remotely sensed precipitation data.

**Figure 5: Probability of failure map detail for a specific time under prevailing rainfall conditions. Known historic landslide initiation points (ellipses) partly overlap with current slope stability conditions. However, high spatial resolution, and therefore a high degree of spatial discontinuity, poses a risk for missing many real landslide events in an early warning situation.**

This spatial confidence buffer that indicates a rather narrow ensemble spread is an equifinal result of the main predetermining

factor: slope angle. Neves Seefelder et al. (2016) and Zieher et al. (2017) identified slope angle as one of the most sensitive modeling parameter in TRIGRS, which is not surprising since slope failures are in general associated with higher slope angles (Liao et al., 2011). Therefore, no matter what the geotechnical or hydraulic input parameters are, it will be always the same slope segments that will result the highest slope failure probability. Slope failure probability will ultimately vary only based on the dynamic component (here: rainfall) or if a spatially distributed soil depth map is provided. The ensemble members in

Fig. 3 indicate very similar results under greatly varying input parameters because of equifinality. This raises the question if model calibration is physically advisable or if we could draw useful conclusions from the direct model output alone (see conclusions *Challenge 1: Parameter uncertainties at regional scale modeling*).

Deterministic forecasts suppress information and judgement about uncertainty. They generally pretend to be absolute based on an optimal set of input parameters. Empirical approaches, such as the commonly used rainfall thresholds in landslide early

warning applications, started to incorporate estimates of uncertainty only recently by defining rainfall thresholds at different exceedance probabilities (e.g. Melillo et al., 2016, Piciullo et al., 2017), yet they rely on very good landslide event catalogues and thus purely on past reportings, which adds a tremendously large source of error (Peres et al., 2017). Gariano et al. (2015) found that an underestimation of only 1% in the number of considered landslides can result in a significant decrease in the performance of a threshold based landslide EWS. Additionally, rainfall thresholds represent a simplification of the underlying

physical processes by establishing purely a relationship between rainfall and landslide occurrence (Bogaard and Greco, 2017). Both, deterministic and empirical approaches may create the illusion of certainty in a user's mind, which can easily lead to wrong conclusions. Krzysztofowicz (2001) mentions a notable event in the spring of 1997, where a falsely issued deterministic forecast on the Red River in Grand Forks, North Dakota, led to evacuations and left a devastated city. After the event a City Council Member in Grand Forks stated (p. 3): "…the National Weather Service continued to predict that the river's crest at





Grand Forks would be 49 ft…If someone had told us that these estimates were not an exact science,…we may have been better prepared."

Based on these words, the presumption that hiding the predictive uncertainty behind the façade of a precise estimate serves better the public need is wrong and careless. Concerns about the acceptance of probabilities in decision making turned out to be unwarranted (Krzysztofowicz, 2001). Based on our observations we found that many published landslide studies dealing with physically based hindcasting applications rely too strong on purely number based validation outputs. Deterministic results are taken as given when the modellers achieve *satisfactory* results based on the model validation, without defining what the criteria are for this satisfaction or when this state of satisfaction is reached. Beven (1996) argues that this is generally owed to relativism, when there is a need to adopt less stringent criteria of acceptability or to acknowledge that it is not possible to predict all the observations all the time (with common arguments ranging from scale issues, spatial heterogeneity, uncertainty in model structure or process understanding, etc.). In all other cases, probabilistic approaches should be prioritized since they allow not only for the incorporation of parametric uncertainties, but also facilitate the geomorphic plausibility control in the absence of proper calibration/validation data. However, narrowing down uncertainties is a good first step, but not the be-all and end-all of ensemble approaches. It is the differences that matter between model predictions and determining and unpicking those differences should be the ultimate goal of ensemble approaches which requires high quality data (Challinor et al., 2014). The scarcity of such high-quality data in landslide research is well known. The potential of local-scale studies to draw conclusions for a larger scale (e.g. Bordoni et al., 2015) remains to be a very important field of study in the near future. In this regard, data assimilation might be a key factor for producing accurate model predictions while reducing those inherent uncertainties. Data assimilation can be referred to as (real-time) parameter updating with observations of flow, soil moisture, groundwater, displacement or rainfall (continuously measured through e.g. radar, rain gauges, etc.) and appropriate uncertainty modeling to correct model predictions (Collier, 2007, Reichle, 2008). Liu et al. (2012) give an in-depth review on the current state of data assimilation applications in both, hydrologic research and operational practices that are in many parts valid for landslide prediction too. While there are a few adaptive systems in landslide early warning based on empirical thresholds (e.g. the SIGMA early warning system in Italy (Martelloni et al., 2012, Segoni et al., 2017)), there are none that use physically based predictions with blends of most recent QPEs or other independent observations. For extreme events, this might be key if the probability of extreme floods or landslides occurring is continuously and objectively evaluated and updated in real-time, especially when it comes to assimilating new observations from multiple sources across a range of spatiotemporal scales (Liu et al., 2012).

## 9 Conclusions

We would like to conclude this paper by raising awareness for a couple of technical and conceptual challenges the landslide forecasting community has to face in the near future. Since physically based, probabilistic landslide forecasting is still in its





infancy, we refrain from addressing challenges in operational practices that are currently discussed in hydrological forecasting (e.g. Pagano et al., 2014), but are of equal importance for the operational use of landslide forecasting nonetheless.

**Challenge 1: Parameter uncertainties at regional scale modeling**

Current practices for geotechnical parametrization in physically based landslide modeling include the application of averaged values from in situ measurements (e.g. Thiebes, 2014, Tofani et al., 2017, Zieher et al., 2017) or using values from existing databases, lookup tables or other published/unpublished sources (e.g. Schmidt et al., 2008, Kuriakose et al., 2009, Mergili et al., 2014b). In the landslide research community, probabilistic treatment of input parameters for regional model application has seen a rise only in the last couple of years. Probabilistic approaches allow for a more thorough consideration of uncertainties and inherent variability of model specific parameters. Spatially varying parameters (both geotechnical and hydraulic) are usually represented as univariate distributions of random variables based on an underlying probability density function and statistical characteristics (Fan et al., 2016). Friction angle and cohesion are commonly considered as such varying variables that are treated in a probabilistic way for model parametrization (e.g. Park et al., 2013, Chen and Zhang, 2014, Raia et al., 2014, Salciarini et al., 2017). Interestingly, in hydrological streamflow prediction the parameter uncertainty of the hydraulic model is often neglected in favor of a deterministic parameter input. This is explained by the superior proportion of total estimation uncertainty introduced by the weather predictions alone, which blurs the streamflow variability that the meteorological input data cannot explain (Alfieri et al., 2012b).

Measuring geotechnical and hydrological parameters for large areas is difficult, time-consuming, and expensive. Therefore, applying spatially distributed physically based models with spatially variable geotechnical parameters is not straightforward and it is impossible to find an approach that is universally accepted (Tofani et al., 2017). Even if there is a sufficiently large amount of measured values available for one, some or even all parameter values in a model up to the point that it is possible to specify distributions and covariances for the parameter values, there remain some methodological obstacles. For example, there is no guarantee that values measured at one scale will reflect the effective values required in the model to achieve satisfactory predictions of observed variables (Beven and Freer, 2001). At larger scales (e.g. > 1:25,000), there are several factors that cause spatial variation of, for example, soil water content, topography, differences in soil depth, -type and -texture, vegetation characteristics, as well as rainfall patterns. Additionally, spatially varying soil and hydraulic properties are influenced by interrelated soil formation processes (such as weathering processes, biological perturbations, atmospheric interactions) (Fan et al., 2016), and thus making selective in situ soil sampling a tricky task when performed at a larger scale. Small scale (e.g. < 1:10,000) variability usually lacks a spatial organization, hence its representation as stochastic process. The larger the scale, however, the more soil forming processes manifest a persistent deterministic signature due to the predetermined geology, topography, climate, etc. (Seyfried and Wilcox, 1995, Fan et al., 2016). Neves Seefelder et al. (2016) suggest applying rather broad ranges of parameters for physically based approaches to be on the "safe side" as they yield results comparable in quality to those derived with best-fit narrow ranges. By acknowledging the fact that geotechnical and hydrological parameters – when applied on a larger scale – are highly variable, uncertain and often poorly known, narrow



parameter ranges or even singular combinations of parameters come with the risk of being off target (Neves Seefelder et al., 2016). This basically implies that, when working at a regional scale and beyond, an actual parametrization with in situ measured samples might not be necessary at all when using literature values instead. This could mean enormous savings in time and money spent, yet this clearly needs further research to evaluate whether there is and to what degree the benefits of

actual sampled in situ data are compared to just utilizing literature values in broad ranges when modeling at larger scales.

**Challenge 2: Rare events and model averaging**

Like flood events, landslides types with a rapid onset can indeed be considered as an extreme event. Hereby, extreme does not necessarily refer to huge displaced landslide volumes – also small landslides might be considered as extreme in terms of potential consequences. While it is possible to continuously monitor and forecast regular streamflow, extreme events are scarce

which makes model calibration and, consequently, forecasting a real challenge. We argue that this is even more so the case for landslides since there are no directly observable target variables to be monitored at a regional scale. Landslide models can only be calibrated on a case by case basis. Shallow landslides are one of the most common landslide types (van Asch et al., 1999). While they occur quite in abundance when looking at their spatial distribution, they are typically low-frequency events. And most of them do occur in so called 'low-risk' environments as defined by Klimeš et al. (2017): low annual frequency of

landslides; the majority of the landslides are of small size and are low impact events. Due to the scarcity of such extreme events, Collier (2007) argues that such events may lie outside of what model calibration if capable of providing for forecasting approaches. Commonly, calibration will improve the reliability of forecasts (i.e. the match of the target variable or forecast probabilities to frequency of observations of the event) but reduce the resolution of the forecast (the ability to discriminate whether an event will occur or not). Consequently, calibration will improve forecasts of common events, but reduces the

probability of forecasting more extreme events.

The WMO (2012) argues that this is the case when events are rare, since the statistical distributions are trained to the more common events. For rare events, hence, calibration cannot be expected to provide significant improvement over the raw forecasts. Therefore, it is very difficult to *validate* a model for future use, as it can be only continually evaluated in the light of the most recent data (Oreskes et al., 1994, Challinor et al., 2014). And landslides are per se extreme events with no *common*

events attributed to them as they only occur under exceptional circumstances given the environmental interactions involved. This raises the question if any averaged model output, and that is by definition every model output based on model calibration from past events, will ever be able to precisely forecast extreme events at the regional scale. The sensitivity of the model had to be lowered in a way that much larger areas of slope failure need to be forecasted to catch a few real extreme events at the cost of significantly raising the number of false alerts. This is especially the case when engineering conservatism comes into

play in decision making, thus leaving probabilistic forecasting attempts in a nonsuperior state over purely deterministic approaches. This is a known issue (e.g. Baum et al., 2010) in a way that FoS computations usually are more likely to identify areas prone to slope failure during a given rainfall event rather than predicting exact locations of specific landslides. A term





such as landslide susceptibility forecasting seems more appropriate in that case. Our results in the Flyschzone of Lower Austria seem to point in that direction so far. This is definitely an issue that needs far more in-depth research in the future.

What else has to be kept in mind are the technical specifications of the modeling approach for slope stability analysis at a regional scale. The most commonly applied modeling approach relies on the infinite-slope stability model which reduces the landslide geometry to a slope-parallel layer of infinite length and width. Modeling approaches that try to introduce more complex landslide geometries in a GIS environment are generally outperformed by the infinite-slope stability model (Zieher et al., 2017). Consequently, parameters representing the landslide geometry assumed by the model (i.e. slope angle and depth) are highly sensitive (Zieher et al., 2017). This means that the underlying model itself already performs some sort of averaging too since the precise landslide geometry cannot be adequately resolved in the infinite-slope stability model.

## Challenge 3: Computational burden

In literature, physically based approaches for modeling rainfall-induced shallow landslides were suggested to be applied to smaller scale study areas while statistical based approaches were recommended for larger scale susceptibility assessments (e.g. van Westen et al., 2006, Corominas et al., 2014). One reason usually mentioned is the poor comprehension of the spatial organization of the geotechnical and hydraulic input parameters (e.g. Tofani et al., 2017, Park et al., 2013). However, as outlined above, it does not make too much difference whether the underlying study area is 50 km² or 5000 km² investigated at a scale of 1:1,000 or 1:25,000 — the model is still influenced by errors or uncertainties from the input parameters to the same degree given the fact how input parameters are derived. Therefore, one major drawback used to be the computational costs involved when modeling physically based at a regional scale. Because as soon as computational power was available at reasonable costs, the area size, associated with a high-resolution DEM, steadily increased over time for physically based applications and currently exceeding thousands of square kilometers (e.g. Tofani et al., 2017, Alvioli and Baum, 2016).

Recent landslide model development is aiming towards featuring multithreading and parallelization. Since high resolution DEM are available in many parts of the world, the computational demands increased significantly, especially when applied in a dynamic/time-dependent modeling framework. Parallelization has great potential in grid-based landslide modeling, especially for the time-consuming hydraulic model components, for several reasons: in case of TRIGRS, for example, which is a coupled slope stability and hydraulic model, only excess water from infiltration is directed to the neighboring cells which makes it the only variable that relies on explicit neighborhood relations. This needs to be done only once, however. Vertical groundwater flow and one-dimensional slope stability in a two-dimensional array of noninteracting columns can subsequently be computed independently for each cell, which is a prime example for parallelization purposes (Baum et al., 2010, Alvioli and Baum, 2016). Besides TRIGRS v2.1, which received its parallel implementation by Alvioli and Baum (2016) only recently, other models for physically based landslide applications are using a parallelized module: NewAge-JGrass (Formetta et al., 2016) or r.slope.stability (Mergili et al., 2014a).

In our case study, the computational time for one model iteration is about 45 minutes, which is far too long for computing a large set of different ensemble members in an operational real-time application. We did not yet use the parallel implementation



of TRIGRS on our regular commodity machine (3.40GHz quad-processor equipped with 32GB RAM), but Alvioli and Baum (2016) reported that parallel computation on a multi-core node already led to a significant speedup compared to a single-node local machine. When applied to a HPC (High-Performance Computing) cluster, they were even able to reduce running time from one day to one hour. This allows the exploration of new possibilities in how landslide forecasting can be approached in

the future. While HPC applications are common in meteorological (Bauer et al., 2015) and hydrological forecasting (Shi et al., 2015), this is a field clearly underexploited in the field of landslide forecasting. This opens up possibilities to encompass fine tuning of input parameters by means of multiple model runs, probabilistic applications and, first and foremost, real-time applications with a continuous consideration of antecedent and forecasted rainfall information (Alvioli and Baum, 2016).

### Acknowledgements

The authors thank the Provincial government of Lower Austria for their support.

Compliance with ethical standards.

The authors declare that they have no conflict of interest.

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





**Figure 1: (A) Location of the Rhenodanubian Flyschzone in Lower Austria (DEM: CC BY 3.0 AT–Federal state of Lower Austria); (B) Typical earth slide in Lower Austria after a heavy rainfall event in May 2014 (Picture: K. Gokesch).**





**Figure 2: Probabilistically derived modeling parameters based on random sampling from a normally distributed state function. Jittering dots (to prevent overplotting) indicate individual samples within a plausible parameter range.**





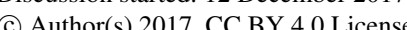

**Figure 3: Postage stamp map for 24 model iterations for the same time. Each ensemble member was initialized with altered parameters within a plausible range to account for variability and spatial uncertainty. Factor of Safety (FoS) values < 1 indicate slope instability.**





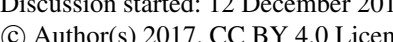

**Figure 4: Probability of Failure depicted as a proportion of the ensemble members that predict an event to occur (FoS < 1.0). Building exposure to current slope failure predictions adds an additional information layer for decision makers. Buildings and roads are imported from the freely accessible OpenStreetMap (OSM) database (© OpenStreetMap contributors).**





**Figure 5: Probability of failure map detail for a specific time under prevailing rainfall conditions. Known historic landslide initiation points (ellipses) partly overlap with current slope stability conditions. However, high spatial resolution, and therefore a high degree of spatial discontinuity, poses a risk for missing many real landslide events in an early warning situation.**

