# Peer review of "Probabilistic landslide ensemble prediction systems: Lessons to be learned from hydrology"

_Natural Hazards and Earth System Sciences, 2017_

## Referee Comment (RC1) · Anonymous Referee #1 · 12 Jan 2018

General comments:

The paper reviews recent developments in applying ensemble prediction systems to probabilistic hydrologic forecasting and uses a case study to demonstrate how ensemble prediction might be applied to landslide forecasts. The paper is well written and could be accepted with minor revisions. The discussion and conclusions would benefit by (1) adding brief remarks about additional sources of uncertainty from DEM data, especially given the sensitivity of the infinite slope model to slope angle; (2) commenting about treatment of large-scale heterogeneity in regional scale model ensembles; (3) clarifying how validating landslide models using extreme events degrades model or forecast accuracy. These points are amplified in my detailed comments below. I have also noted several minor editorial corrections.

[Figure]

Specific comments:

Page 7, lines 5-7, Greco and Pagano (2017) seem to indicate in their Figure 2, that warning needs to start during the latter part (triggering rainfall) of Phase 1. Common sense indicates that warning or at least issuing an advisory that slides are likely with additional rainfall during this stage is prudent. Waiting until stage II is probably too late.

Page 13, line 1, it is probably worth mentioning either here or somewhere that for the sake of simplicity, you varied only three of the most sensitive model parameters, cohesion, friction, and soil depth, but in an operational landslide forecasting system varying additional sensitive parameters would be prudent.

Page 15, line 19, consider inserting "or a property zone map" between "soil depth map" and "is provided"

Page 16, line 14, What is meant by "unpicking?" separating?

Page 17, lines 23-33, Although "large scale" and "small scale" are used correctly in previous sections of the paper, they are used incorrectly here. Large scale maps and models are detailed (see https://en.wiktionary.org/wiki/large-scale). Similarly, small-scale maps and models are generalized (cover large area with little detail). It would be clearer to use "local scale" and "regional scale" or similar terms.

Page 17, lines 25-34, occurrence of landslides on more gentle slopes that have low susceptibility according to the model ensemble (Fig. 5, Page 14, lines 28-29,) as well as the modeling artifact mentioned on page 14, lines 9-10 suggest that even using an ensemble, the modeler needs to account for certain inhomogeneity, including infrastructure, such as retaining walls, as well as broad deterministic differences imposed by geology, etc. Given the high sensitivity of the infinite slope model to slope, why are effects of uncertainty in the DEM on model results still mostly unexplored? Shouldn't DEM uncertainty be an area for further research?

Page 18, lines 3-5, See Gioia et al. (2015) for a case study of using literature values to

parameterize such a model.

Page 18, lines 21-29, The WMO (2012) argument makes sense for meteorological and hydrological models because they are calibrated to variables that can be measured continuously (temperature, humidity, precipitation, streamflow, and so on). Please clarify how the argument applies to landslides when the models are calibrated to landslide events, which as the authors point out, are rare. Thus, in this case, the statistical distributions are trained to the extreme events. The only common, almost daily events to which landslide models could be compared for validation are the absence of landslides. While it's true that most recent publications about rainfall thresholds for landslides have included some non-landslide inducing precipitation in developing or validating thresholds; it seems uncommon to continuously evaluate those thresholds against daily absence of landslides. This is an important point, because, if I understand correctly, you are arguing that the way that process-based landslide susceptibility models are being calibrated or validated (by comparison with past events) is biasing them in a way that decreases forecast accuracy. If so, and a new or different approach is needed for validating models, what do you suggest?

Page 18, line 27-29, The problem here is not just a matter of lowering model sensitivity. In many cases, available data are not adequate to create a very sensitive model using any procedure. For example, in the rolling hill terrain of the Esino River Basin, Gioia et al. (2015) found there was no well-defined relationship between topographic variables and landslides, making it difficult to attain high model sensitivity. For other areas, such as the Colorado Front Range (Alvioli and Baum, 2016), model sensitivity was impaired by the quality of available DEMs, such as those derived from legacy photogrammetrically mapped topographic contour data, as well as other data uncertainties. Mergili et al. (2014b) similarly seem to have experienced difficulties obtaining high sensitivity even when using a 3-D method and ranges of depth, cohesion and friction. While I appreciate main the point of your paper that accounting for uncertainty through the use of model ensembles will improve forecasts, it seems clear that continued research is

needed on other fronts as well to overcome some of the challenges in making accurate forecasts.

Page 19, lines 8-9, What do you mean here by averaging performed by the infinite-slope stability model? Don't 3-D models such as the model of Xie et al. (2003, 2004, 2006), r.slope.stability (Mergili et al. 2014a), and Scoops3d (Reid et al. 2015) perform a sort of averaging over the neighborhood of each point in the search grid? By considering a group of neighboring cells in each trial failure, the 3-D models effectively average out effects of the ground surface irregularities at the same time as they account for effects of the finite extent and lateral boundary effects of realistically shaped trial landslides.

Technical corrections:

Page 2, line 29, change "explicitly introduces in into the model" to "explicitly introduces it into the model"

Page 6, line 24, capitalize "it"

Page 8, line 22, change "provide" to "provides"

Page 8, line 31, change "if" at the end of the line to "it"

Page 13, line 13, change "a proof on concept" to "a proof of concept"

Page 15, line 16, change "parameter" to "parameters"

Page 16, line 22, delete comma after "data assimilation applications in both"

Page 18, line 16, change "calibration if capable" to "calibration is capable"

Page 18, line 27-28, change "had to be" to "must be"

Page 18, line 30, change "a nonsuperior state over" to "an inferior state below"

Figure 4, add outline of area shown in Figure 5.

References cited:

Alvioli, M. and Baum, R. L.: Parallelization of the TRIGRS model for rainfall-induced landslides using the message passing interface, Environmental Modelling & Software, 81, 122–135, doi:10.1016/j.envsoft.2016.04.002, 2016.

Gioia, E., Speranza, G., Ferretti, M., Godt, J.W., Baum, R.L., Marincioni, F., 2015, Application of a process-based shallow landslide hazard model over a broad area in Central Italy: Landslides, doi: 10.1007/s10346-015-0670-6, p. 1-18.

Greco, R. and Pagano, L.: Basic features of the predictive tools of early warning systems for water-related natural hazards: examples for shallow landslides, Natural Hazards and Earth System Sciences Discussions, 1–31, doi:10.5194/nhess- 2017-269, 2017.

Mergili, M., Marchesini, I., Alvioli, M., Metz, M., Schneider-Muntau, B., Rossi, M. and Guzzetti, F.: A strategy for GIS-based 3-D slope stability modelling over large areas, Geoscientific Model Development, 7(6), 2969–2982, doi:10.5194/gmd-7- 2969-2014, 2014a.

Mergili, M., Marchesini, I., Rossi, M., Guzzetti, F. and Fellin, W.: Spatially distributed three-dimensional slope stability modelling in a raster GIS, Geomorphology, 206, 178–195, doi:10.1016/j.geomorph.2013.10.008, 2014b.

Reid ME, Christian SB, Brien DL, Henderson ST (2015) Scoops3D—software to analyze three-dimensional slope stability throughout a digital landscape. Virginia: U.S. Geological Survey Techniques and Methods 14-A1, 218 p., https://dx.doi.org/10.3133/tm14A1

WMO: Guidelines on Ensemble Prediction Systems and Forecasting, World Meteorological Organization, WMO-No. 1091, Geneva. Available at: http://www.wmo.int/pages/prog/www/Documents/1091_en.pdf, last access: 30 November 2017, 23 pp., 2012.

[Figure]

Xie, M., Esaki, T., Zhou, G., and Mitani, Y.: Three-dimensional stability evaluation of landslides and a sliding process simulation using a new geographic information systems component, Environ. Geol., 43, 503–512, 2003.

Xie, M., Esaki, T., and Zhou, G.: GIS-based Probabilistic Mapping of Landslide Hazard Using a Three-Dimensional Deterministic Model, Nat. Hazards, 33, 265–282, 2004.

Xie, M., Esaki, T., Qiu, C., and Wang, C.: Geographical information system-based computational implementation and application of spatial three-dimensional slope stability analysis, Comput. Geotech., 33, 260–274, 2006.

---

## Referee Comment (RC2) · Anonymous Referee #2 · 13 Jan 2018

General comments

Landslide early warning is a very interesting and relevant topic. However, in my view the authors fail to deliver what they promise. After reading a 10 page long introduction, I was convinced that the authors were going to introduce a new framework/method to incorporate ensembles of weather forecasting for landslide prediction, similarly to what has been done for hydrological forecasting. However, that is not the case. The authors selected a landslide physically based model, which they run considering uncertainty in some pre-selected model parameters. Nowhere, did the authors considered uncertainty introduced by the weather forecast that is essential in early warning. While it is valid to use a Monte Carlo approach to evaluate uncertainty introduced by the parameters of the landslide model (many studies have done this, both in the hydrology

and the landslide community), this is different from assessing uncertainty in a weather forecast (by using an ensemble for weather forecasting, provided by operational forecasting system such as those listed on page 6, lines 12-17) particularly relevant for early warning systems. So my main concern with this paper is that while the reader is given the impression that the paper is about uncertainty in weather forecasts to improve landslide early warning systems, the paper is about something else, i.e. uncertainty in landslide model parameterisation. Why then writing so much about early warning and ensembles for weather forecasting (one example is the entire section 4, but I will provide more examples in the Specific Comments section) if the paper is about something else? I believe that this study could be a really interesting if the authors have indeed used landslide physically based models together with many different initial conditions of rainfall (based on a possible weather forecast), to enhance a landslide early warning system at a regional scale. Given the results presented, in my opinion this paper needs to be restructured so that the first six sections reflect what the paper is really about, that is uncertainty related to the geotechnical parameterisation of the landslide model, as stated on page 10, line 5.

Other major comments include:

- Clearly state what is novel about this paper and its direct relevance for early warning. Monte Carlo simulation as a way to incorporate parameter uncertainty is not new (just few examples in a landslide context, among many, Haneberg, 2004; Cho, 2007; Melchiorre and Frattini, 2012).

- Lack of justification on why only certain model parameters were considered uncertain and not others.

- Not enough detail is provided on the probability distributions used for the uncertain parameters, which is essential for the reproduction of the results.

- The paper lacks a proper structure. It seems more like a thesis chapter, than a paper for a journal. The Introduction is too long (six separate sections), the discussion is

incomplete, and the main conclusions that arise from this study are non-existent. What the authors call Conclusions section, reads more like a Discussion section. The fact that the study lacks a Method section, jumping straight from Introduction to Case Study, makes the paper more difficult to read.

Specific comments

Page 1, line 13: What do the authors mean by "larger scales"?

Page 1, line 14: Why convective-scale numerical weather predictions specifically and not other numerical weather predictions? Please also briefly explain what convective-scale numerical predictions are.

Page 1, lines 17-20: Is this study going to answer any of the future research directions identified by the authors in the Abstract? Or are the future research directions identified here, gaps that need to be addressed in the future? If the first, please make clear in the Abstract what the main conclusions of this study are and how this study contributes to the points raised. If the latter, before pointing the future directions of research, please make clear what the main conclusions and contributions of this study are.

Page 1, line 31 – Page 2, lines 1-2: The authors say that "For natural hazard types with a rapid onset (. . .) rainfall can be considered as the main triggering mechanism." What about earthquake triggered landslides? They also have a rapid onset and are not triggered by rainfall. Or wildfires, etc.?

Page 2, lines 7-12: The authors say on page 2 that there are multiple reasons why numerical weather predictions are not used within the landslide early warning community. One of those reasons pointed by the authors is the "the complexity of single landslide detachments: the same landslide triggering event does not necessarily cause other landslides as the time between propagation stage and the collapse phase may vary significantly based on differences in local conditions (. . .) and spans from minutes (. . .) to years (. . .)". While it is true that the same rainfall event may result in a landslide

in certain places (and for certain initial conditions) but not in others, and that the collapse phase may vary from minutes to years, I do not fully understand how this relates to the lack of uptake of numerical weather predictions by the landslide early warning community. Please make the link clearer to the reader.

Page 2, lines 25-29: The authors claim that "This paper reviews and summarizes concepts of ensemble prediction systems (EPS) in hydrology and how those can be translated to be applicable also in process-based landslide early warning systems. A strong emphasis is put on how to deal with spatial uncertainties by demonstrating the benefits of probabilistic model application which does not eliminate uncertainty, but it explicitly introduces in into the model results." While the authors provide a case study where they consider spatial uncertainties and how these uncertainties impact the model output (Factor of Safety), I do not agree that the authors show how ensemble predictions systems can be used in process-based landslide early warning systems. For early warning systems, it is essential to consider the uncertainty in weather forecasts, and not just the uncertainties introduced by model input parameters as the authors do in their case study. Please make the text here more accurate, so that it reflects what is later delivered.

Page 2, lines 31-33, Page 3, lines 1-2: The authors state at the end of Section 1, that the aims of the paper are: "a) to critically evaluate the current state of physically based landslide early warning, its limitations and possible ties to hydrological forecasting; b) on this basis, to foster cooperation across disciplinary boundaries to bring together scientists from different fields to pursue research based on forecasting experiences gained in the last couple of years". Is this paper a literature review on this topic, or does it aim to introduce something new? If the first, please make it clear. If the latter, please clarify what are the research questions that the authors aim to answer.

Page 3, lines 29-30: Streamflow measurements for extremely high conditions are difficult to obtain and far from accurate. Please consider rephrasing the sentence in your manuscript.

Page 4, lines 10-11: How does this sentence link to what comes before? Why do the authors jump to data collection?

Page 4, lines 12-25: Why do the authors talk about underestimation of landslide losses here? Why is this relevant for the overall argument that the authors making? Please clarify.

Page 4, lines 27-28: Perturbations to model parameters is one thing, perturbation to starting/initial conditions is another (as it is clear from the definition given by WMO that the authors quote in page 3, lines 5-11). It is crucial to clarify what this study is about. Up to this point, the reader is made to believe that perturbed initial conditions (reflecting uncertainty in weather forecasts) are going to be used (and maybe also different combinations of physical parameterisation schemes of the landslide model). However, that is not the case. The study is about uncertainty in the physical parameterisation of the landslide model. Please note, that I acknowledge that the authors make a more clear distinction later on page 5, lines 5-9, when they define the term "ensemble prediction" for multi-parameter and multi-model predictions. But my point is that, the reader does not know what the paper is about. And if the paper is going to be about parameter uncertainty (not clear at this point), why do the authors spend so much time talking about forecasting, weather ensemble forecasting and early warning?

Page 5, line 10: I do not agree that there have been only a few attempts to use ensemble techniques in landslide research (assuming here that the authors are talking about multi-parameter and multi-model ensembles). Some examples include: Haneberg, 2004; Rubio et al, 2004; Cho, 2007; Melchiorre and Frattini, 2012, Arnone et al, 2014. Some of the references provided by the authors later (page 5, line 27) are possibly also good examples.

Page 5, line 12: Once again when the authors say that "None of them, however, incorporate ensemble techniques in real-time application", makes me believe that this study will address that, what in my opinion is not the case.

Page 5, lines 28-32: Similarly to the previous comment, when the authors say that "Haneberg (2004), Park et al. (2013), Raia et al. (2014), Lee and Park (2016) and Zhang et al.(2016) treat soil properties at regional scale applications in a probabilistic way by randomly selecting variables from a given probability density function, mostly by means of Monte Carlo (MC) simulation (...) None of those probabilistic approaches are operated in spatial real-time-early warning systems, not even on a prototype basis", makes me believe that this study will go beyond using Monte Carlo simulations of soil properties randomly sampled, and provide a method/case study/framework to be used it in real time warning systems. That does not turn out to be the case.

Page 6, line 28: What do the authors mean by hydrological applications?

Page 7, lines 8-26: Again, why do the authors spend so much time talking about quantitative precipitation forecasts (QPF) from numerical weather predictions, including giving the example of flash floods that require QPFs with 1-6 hour lead times, if the study does not use such QPFs?

Page 8, lines 5-6: This is a confusing sentence mixing equifinality and nonlinearity. Please consider rephrasing it.

Page 8, line 31: The concept Factor of safety is introduced here (page 8, line 31), and defined later on page 11, lines 21-24.

Page 9, lines 11-13: The sentence "Commonly, calibration will improve the reliability of forecasts (i.e. the match of the target variable or forecast probabilities to frequency of observations of the event) but reduce the resolution of the forecast (the ability to discriminate whether an event will occur or not)." appears twice in this manuscript (page 9, lines 11-13, and page 18, lines 17-19). Please do not repeat the exact same text twice in the same manuscript. It is also unclear what the authors exactly mean. I have not seen before "resolution of forecast" being defined as "the ability to discriminate whether an event will occur or not"

Page 9, lines 13-14: "calibration will improve forecasts of common events, but will also lead to the underprediction of more extreme events."- Please add reference(s).

Page 9, lines 17-26: This paragraph is confusing. At the beginning of the paragraph the reader is given the impression that the authors are going to talk about validation. But then the paragraph seems to be about calibration.

Page 9, lines 24-25: What do the authors mean by "dependence on the model structure"?

Page 9, line 33 and page 10, lines 1-2: What is the point of providing these two references here, without a brief description of the measures? If those measures are relevant for the case study, please consider giving a brief description here. If not, I am not sure why the authors introduce the references here.

Page 10, line 3: After 10 pages of introduction (not sure why it has to be so long!), it would be helpful to state clearly what are the challenges that are going to be addressed in this paper and the research questions, before moving to the case study section. So far, this reads more like a book/thesis than a journal paper. Jumping from introduction to case study is also not very common.

Page 10, line 4: What do the authors mean by "simplified ensemble modelling"? What has been simplified?

Page 10, lines 5-7: It is unclear what is done in this study regarding point c) "how infrastructure data can further supplement early warning procedures in an exposure context." Later on in the study (page 14, lines 3-5), the authors overlap the results of the landslide model for a past rainfall event with Open Street Map. However, given that this is based on a past rainfall event (and not forecasting weather) how can this be used to supplement early warning procedures? Please rephrase point c) to reflect the results shown later on, or consider deleting point c) from this list.

In here the authors state more clearly what the aims of this case study are. Taking into

account that uncertainty in weather forecasts in landslide prediction is not part of those aims, why spending so much text up to here on that that topic?

Page 10, lines 26-27: Please add references after "physically based models can be quite commonly found to evaluate rainfall-induced landslide susceptibility at the regional scale"

Page 12, lines 4-7: The description of the model setup is too generic. For example, what is the time step used? How many parameters does the model have?

Page 12, lines 4-7: What are the properties of the normal distribution that the authors sample from (i.e. mean and standard deviation) and how were they derived? On what grounds did the authors select a normal distribution and not any other distribution?

Page 12, lines 7: Why 25 model runs? Such a small number may not adequately represent the parameter space.

Page 12, line 8: What is the "initial model run"? Is it the first of the 25 model runs?

Page 12, lines 9-12: Please consider rephrasing this sentence, as it is confusing. The authors say that the probability of failure of a given cell is equal to dividing the number of unstable raster cells by the number of model runs. It is confusing as for "this raster cell" they count the "number of unstable raster cells". I suppose that the authors mean the number of simulations that lead to FoS<1 for that specific raster cell, divided by the total number of simulations (i.e. 25). Is that the case? If so, please change the text to reflect that.

Page 12, line 21: Is the data from Tofani et al. (2017) mentioned here used by the authors to derive the parameters of the probability distributions of the uncertain parameters (i.e. mean and standard deviation of the normal distributions)? If so, and as mentioned in a previous comment, please provide the values of mean and standard deviation of the distributions and how those values were determined.

Page 12, lines 25-26: "(. . .) the boxplots suggested normal to lognormal parameter

distributions. This is a common observation and might be a result of the central limit theorem". A lognormal distribution is not the result of the central limit theorem.

Page 12, line 25: If the boxplots suggest normal to lognormal distributions, why did the authors decide on normal distributions? Please justify the choice made. Is the observation "the boxplots suggested normal to lognormal parameter distributions" valid for all model parameters? Please detail which parameters show a normal distribution and which parameters show a lognormal distribution.

Page 12, line 30: Why do the authors introduce GLUE here? GLUE indeed involves Monte Carlo simulation, but there is more to GLUE than that. In this study, Monte Carlo simulations are performed, but GLUE is not. So I do not see the need to mention GLUE here. It may only contribute to confuse readers that are not familiar with GLUE.

Page 12, lines 31-32: Are the parameters sampled from a predefined parameter range or are they sampled from normal distributions? Please clarify. Page 12, lines 1-2 and line 33: Please justify why only uncertainties related to soil depth, effective cohesion and effective friction angle were considered. How do the authors know what are the most influential model parameters, without having carried out a sensitivity analysis?

Page 13, line 6 (caption of figure 2): What is "state" function?

Figure 2 (page 27): As mentioned in a previous comment, please provide the parameters of the normal distributions. Some of the points in Figure 2 do not seem to come from a normal distribution (e.g. for friction angle the distance between the 25th percentile and 50th percentile is quite different from the distance between the 50th percentile and 75th percentile).

Page 12, line 7 and page 13, line 11: Is the model run for a specific rainfall event? Why 3 hourly time steps? Was 3 hours the duration of the rainfall event? Please provide enough detail on the rainfall event used to run the landslide model and why that specific event has been selected. Page 13, lines 13-14 and lines 20-21: This is where the study

falls short, i.e. show how numerical weather predictions and related uncertainty could be incorporated in a real-time application. If the landslide model is run with rainfall over the last three hours (page 13, lines 20-21), instead of a rainfall forecast, this cannot be used for early warning. But wasn't early warning the main selling point of this study?

Page 13, line 20: Are there 24 or 25 model simulations?

Figure 3 (page 28) and page 13, lines 23-25: The 24 figures are not very helpful, as the reader cannot see the differences between the maps. What is the message that figure 3 is trying to illustrate? Is it essential to show these 24 maps, given that the reader cannot see much?

Based on the figures provided I cannot see whether "the results indicate quite significant changes across individual members, but also quite high similarities although parameters change drastically between some of the members" (page 13, lines 23-25)

Page 13, line 32: Please make clear what uncertainties are accounted for, given that only certain model parameters were considered uncertain, and other uncertainties such as model structural uncertainty and rainfall uncertainty, were not considered.

Page 14, line 2: What do the authors mean by "this specific time"? Please clarify.

Page 14, lines 7-8 and figure 4 (page 29): The results shown in this study refer to a specific rainfall event. This should be made clear, namely in the main text and in the captions of figures 4 and 5. If it does not rain, the values shown in figures 4 and 5 are no longer the probability of failure. And if it rains a lot (meaning much more than the rainfall event used to run the landslide model) the probability of failure is much higher than what is shown in the figures. Please make clear what the probability of failure refers to.

Page 14, line 11: It is incorrect to say that a narrow ensemble spread is an expression of equifinality.

Page 14, lines 12-13: There is not enough evidence in the results shown in this

manuscript to make such a statement, as the authors did not run the model for different rainfall events. How can the authors know that for a different rainfall forcing the location of possible slope failures is not different?

Page 14, lines 28-29: Indeed, but there are equally also many landslide initiation points that do not correspond to high failure probability. Of course, this may have to do with the rainfall input used. But that is my point - the probabilities shown in the map only have meaning for the specific rainfall event used, which the reader does not even know what it was.

Page 14, line 32: "there are still many accounted uncertainties": Please list the unaccounted uncertainties (or at least some of them).

Page 15, line 6: Why are some real landslides missed?

Page 15, lines 14-15: What do the authors exactly mean by "spatial confidence buffer"? Are these the coloured areas in figures 4 and 5? How does the "spatial confidence buffer" show a narrow ensemble of spread? Spread of what? How does that relate to an equifinal result? How does that relate to slope angle?

Page 15, lines 14-14: How do the authors know that slope angle is the main predetermining factor based on their results? No sensitivity analysis has been performed, to make such a statement. If this statement is based on other studies, that needs to be made clear. It is important to highlight that the most influential parameter highly depends on the probability distributions used.

Page 15, lines 17-18: The statement "no matter what the geotechnical or hydraulic input parameters are, it will be always the same slope segments that will result the highest slope failure probability" is not accurate. Geotechnical and hydraulic input parameters may still matter, even if slope angle may have the greatest impact on the model results. A slope with a certain angle (and assuming the same rainfall event), may fail for a certain combination of geotechnical and hydraulic parameters and not fail

for another combination of geotechnical and hydraulic parameters.

Page 15, lines 18-19: It is unclear what the sentence "Slope failure probability will ultimately vary only based on the dynamic component (here: rainfall) or if a spatially distributed soil depth map is provided" mean. Does failure of probability only depend on rainfall and soil depth map? What about the other aspects, such as slope angle, friction angle, cohesion etc?

Page 15, lines 26-27: How does that add large errors? Please expand. How does that compare to the approach introduced in this paper, where only a pre-selected rainfall event has been used? Page 16, lines 13-14: What does the sentence "However, narrowing down uncertainties is a good first step, but not the be-all and end-all of ensemble approaches." mean? What does it mean "be-all and end-all of ensemble approaches"? How do the authors suggest narrowing down uncertainties? Page 16, lines 23-25: But the authors do not do use physically based predictions with blends of most recent quantitative precipitation estimates either. The authors fail to show how the approach/results they present in this study can be used for early warning. For example, could the model be run fast enough to be used for early warning based on the weather predictions? This is just a very simple example.

Pages 16 and 20, Conclusions section: Line 30: The Conclusions section must start by clearly stating what was learnt from this study, i.e. what the actual conclusions of this study are. Only after this, the authors should discuss any challenges and/or future direction.

Large parts of the Conclusions section should be moved into the Discussion section.

Page 17, line 8: Please provide references after "(. . .) probabilistic treatment of input parameters for regional model application has seen a rise only in the last couple of years."

Page 18, lines 2-3: "when using literature values instead": It is unclear what the authors

mean. Are the authors saying that literature values should be used instead, and that sample measures should be discarded? Please clarify.

Page 19, lines 15-17: I am not sure that this sentence makes sense. Furthermore, the results presented in this study do not allow the authors to make such a strong statement.

I would expect that if we are looking into a smaller area, the variability of certain soil properties (or more generally input parameters) is smaller than if we would be measuring the same properties across a larger area. But most importantly, the authors do not show results to back up their statement.

Page 19, line 20: I do not follow the argument. According to the authors, is computation problem still a problem nowadays or not?

Page 20, lines 5-6: People have been using HPC in landslide modelling – please check the literature.

Technical corrections

Page 1, line 16 – "how ties to. . ." – Please rephrase.

Page 2, line 19: "Another reason for the negligence of physically based forecasting initiatives (. . .)" – Please rephrase

Page 2, lines 24-25: "The hydrological community has recently adopted to those advancements (. . .)" – Please rephrase.

Page 2, line 29: "in into" – Please correct.

Page 3, line 26: In the sentence "One reason why landslide forecasting is seemingly more challenging can be (. . .)", please state what landslide forecasting is more challenging compared to, i.e. "One reason why landslide forecasting is seemingly more challenging THAN X can be (..)"

[Figure]

Page 6, line 13: Delete the comma after "In".

Page 7, line 16: "longer lead times (. . .)" Longer relative to what?

Page 8, line 31: "(. . .) when if" – please correct.

Page 9, lines 17-18: What do the authors mean by "either-or-situations"?

Page 14, line 30: What situation are the authors referring to? Please clarify.

Page 14, line 2: Please clarify what "This" refers to, or in other others explain what is quite detrimental. Is it explicitly accounting for uncertainty?

References

Arnone, E., Dialynas, Y. G., Noto, L. V. and Bras, R. L. (2014). Parameter uncertainty in shallow rainfall-triggered landslide modeling at basin scale: A probabilistic approach. Procedia Earth and Planetary Science, 9, 101-111, doi: 10.1016/j.proeps.2014.06.003.

Cho, S. E. (2007). Effects of spatial variability of soil properties on slope stability, Engineering Geology, 92(3-4), 97-109, doi: 10.1016/j.enggeo.2007.03.006.

Haneberg, W. C. (2004). A rational probabilistic method for spatially distributed landslide hazard assessment, Environmental & Engineering Geoscience, 10(1), 27-43, doi: 10.2113/10.1.27

Melchiorre, C. and Frattini, P. (2012). Modelling probability of rainfall induced shallow landslides in a changing climate, Otta, Central Norway, Climatic Change, 113, 413–436, doi:10.1007/s10584-011-0325-0.

Rubio, E., Hall, J. W. and Anderson, M. G. (2004). Uncertainty analysis in a slope hydrology and stability model using probabilistic and imprecise information. Computers and Geotechnics, 31(7), 529-536. doi: 10.1016/j.compgeo.2004.09.002

---

## Referee Comment (RC3) · Anonymous Referee #3 · 17 Jan 2018

GENERAL COMMENT

The article "Probabilistic landslide ensemble prediction systems: Lessons to be learned from hydrology" presents an analysis of ensemble prediction systems in order to apply them in the probabilistic prediction of landslide occurrence.

The paper is very long, especially in the introduction. However, it is well written, in a good English language. It follows somehow the IMRaD structure, even with some drawbacks, that should be improved. The subject is within the topic of special issue of NHESS journal.

In my opinion, the manuscript needs major revisions before being accepted for publication.

Mainly, the theoretical background described in the introduction is extremely long! I suggest a strong revision of this part aimed at shortening it. The same is for the conclusions section, which can be shortened. There are some parts of the introduction that should be moved in the discussion.

On the other hand, the description of the method used for validating the background is quite fast, as for the results and discussion. Moreover, it seems that the Authors applied a method different from that extensively presented and discussed in the introduction. All these issues should be addressed in the revised version of the paper.

SPECIFIC COMMENTS

At page 1, lines 28-30, Authors state: "In this paper, we use prediction systems synonymously with early warning systems for terminological consistency within the landslide community although we acknowledge that early warning should also cover dissemination and response strategies". I strongly disagree with this terminological association. As acknowledged, an early warning system includes a prediction system and many other components. Thus, if the "landslide community" has used the two terms as synonyms since now, this paper could be a milestone in proposing a separation between them. I suggest to distinct the two items.

Page 3, lines 23-24: Also, Intrieri et al. (2013) have presented a complete scheme for landslide early warning systems.

Page 4, lines 13-14: Please give some examples of prototypal landslide early warning systems.

Page 7, lines 1-3: I think that the time between triggering/propagation and collapse stages varies also according to the landslide types.

Section 7.2.1 is a description of a model, thus it should not be in the "Case study" section.

Page 11, lines 18-19: Recently, Tran et al. (2017) proposed an application of TRIGRS

with a 3D model to analyze 3D slope stability.

At page 12, line 7, Authors state that they computed 25 model runs. At page 13, line 20, they refer to 24 model iterations. Please explain.

Please check all the brackets in the text: somewhere, in particular in relation to references, there are many of them.

Please check the acronyms in the text. Use always acronyms after defining them.

FIGURES

Figure 1.

I suggest to add a map of the whole Austria with the indication of Lower Austria (for non-European readers).

I can't understand the meaning of the elevation classes. I suggest using a continuous scale.

Figure 3.

This figure is extremely hard to see and read. It's quite impossible to see the differences between the different maps. I suggest to split it in 2 or 3 figures or to leave in the text just 6 or 9 significant cases and to put the other maps in an ancillary file.

Figure 4.

There are incongruities in the legends of "probability of failure" and "building exposure". As an example, a pixel with probability of failure (or building exposure) equal to 0.25 is in the first or in the second class? The same for values equal to 0.50 and 0.75. Please correct them including or excluding the extremes in the classes as appropriate.

Why the forested areas are reported in the map?

Figure 5.

As for the previous figure, I suggest to correct the incongruities in the legends and to explain why the forested areas are reported in the map.

I can't understand the symbols used for indicating landslide head scarps. If two dimensions are not needed, I suggest using a point layer.

TECHNICAL CORRECTIONS

Page 3, line 14: please delete "p. 1".

Page 3, line 16: please delete "p. 1".

Page 9, line 1: please correct "this event".

Page 10, line 16: please correct "km2".

Page 10, line 18: please change "," into ";" in the references.

Page 13, line 10: insert "of" in "representation surface topography".

Page 16, line 30: Replace "a couple" with "some".

Page 18, line 7: correct "landslides types".

REFERENCES

Intrieri, E., Gigli, G., Casagli, N., and Nadim, F.: Brief communication "Landslide Early Warning System: toolbox and general concepts". Nat. Hazards Earth Syst. Sci. 13, 85–90, doi:10.5194/nhess-13-85-2013, 2013.

Tran, T.V., Alvioli, M., Lee, G., and An, H.U.: Three-dimensional, time-dependent modeling of rainfall-induced landslides over a digital landscape: a case study. Landslides, doi:10.1007/s10346-017-0931-7, 2017.

---

## Author Comment (AC1) · 7 May 2018

Dear referee #1, dear editor. First of all, we would like to apologize that our replies to your comments are somewhat late. There have been changes of positions and countries of residence among the authors. In order to facilitate a more timely revision we have now invited a new author to the team to support the revision and finalization of the manuscript. We also want to thank you for your thoughtful comments on our manuscript. For our revision, we intend to closely follow your and the other reviewers' recommendations. Please find our replies to your specific comments below. Kind regards, The authors

Anonymous Referee #1 General comments: The paper reviews recent developments

in applying ensemble prediction systems to probabilistic hydrologic forecasting and uses a case study to demonstrate how ensemble prediction might be applied to landslide forecasts. The paper is well written and could be accepted with minor revisions. The discussion and conclusions would benefit by (1) adding brief remarks about additional sources of uncertainty from DEM data, especially given the sensitivity of the infinite slope model to slope angle; (2) commenting about treatment of large-scale heterogeneity in regional scale model ensembles; (3) clarifying how validating landslide models using extreme events degrades model or forecast accuracy. These points are amplified in my detailed comments below. I have also noted several minor editorial corrections.

REPLY: Thank you. We will discuss the respective topics (DEM uncertainties, spatial heterogeneity, and extreme event validation) in our revised manuscript.

Specific comments: Page 7, lines 5-7, Greco and Pagano (2017) seem to indicate in their Figure 2, that warning needs to start during the latter part (triggering rainfall) of Phase 1. Common sense indicates that warning or at least issuing an advisory that slides are likely with additional rainfall during this stage is prudent. Waiting until stage II is probably too late.

REPLY:We agree that warnings or advisory information during the later stages of a landslide triggering rainfall event might already be too late to initiate effective counter measures (e.g. evacuations). In our manuscript, we therefore formulated the sentence to indicate that "warnings should generally be issued during indications of stage (II)" (i.e. the onset of a potentially triggering rainfall event). In any case, the provision of warnings and even landslide outlooks is a highly sensitive issue which must be carried out by an authority with the legal and political responsibility and within the existing legal framework.

Page 13, line 1, it is probably worth mentioning either here or somewhere that for the sake of simplicity, you varied only three of the most sensitive model parameters,
cohesion, friction, and soil depth, but in an operational landslide forecasting system varying additional sensitive parameters would be prudent.

REPLY: We will add this information as suggested.

Page 15, line 19, consider inserting "or a property zone map" between "soil depth map" and "is provided"

REPLY: The integration of a property zone map would not influence the failure probability map but would indeed affect the risk as well as the decision on a warning.

Page 16, line 14, What is meant by "unpicking?" separating?

REPLY: We meant "unpicking" in the sense of untangling. Specifically in this sentence we want to say that the differences between model predictions matter and that it should be the goal to explore the reasons for the differences. For our revision, we intend to use the word "exploring" instead of "unpicking" to avoid misunderstandings.

Page 17, lines 23-33, Although "large scale" and "small scale" are used correctly in previous sections of the paper, they are used incorrectly here. Large scale maps and models are detailed (see https://en.wiktionary.org/wiki/large-scale). Similarly, small scale maps and models are generalized (cover large area with little detail). It would be clearer to use "local scale" and "regional scale" or similar terms.

REPLY: Yes, we were not consistent with the usage of large and small scales. In our revised manuscript, we will correct this and preferably use terms which are less prone to be confused (e.g. large area, regional scale).

Page 17, lines 25-34, occurrence of landslides on more gentle slopes that have low susceptibility according to the model ensemble (Fig. 5, Page 14, lines 28-29,) as well as the modeling artifact mentioned on page 14, lines 9-10 suggest that even using an ensemble, the modeler needs to account for certain inhomogeneity, including infrastructure, such as retaining walls, as well as broad deterministic differences imposed by geology, etc. Given the high sensitivity of the infinite slope model to slope, why are

NHESSD
effects of uncertainty in the DEM on model results still mostly unexplored? Shouldn't DEM uncertainty be an area for further research?

REPLY: Of course, DEMs are not perfect and necessarily represent a simplified topography. We think that since the advent of high resolution LIDAR DEMs, there is not much that researchers and practitioners could complain about in relation to DEM quality. But we agree that the modeler must take into account errors and artifacts present in the DEM (e.g. in our case the retaining wall which has an unrealistically high probability of failure). For a real-time application of our landslide forecasting model, such areas would need to be excluded. In any case, we intend to also discuss DEM related uncertainties in our revised manuscript.

Page 18, lines 3-5, See Gioia et al. (2015) for a case study of using literature values to parameterize such a model.

REPLY: Thanks for the hint.

Page 18, lines 21-29, The WMO (2012) argument makes sense for meteorological and hydrological models because they are calibrated to variables that can be measured continuously (temperature, humidity, precipitation, streamflow, and so on). Please clarify how the argument applies to landslides when the models are calibrated to landslide events, which as the authors point out, are rare. Thus, in this case, the statistical distributions are trained to the extreme events. The only common, almost daily events to which landslide models could be compared for validation are the absence of landslides. While it's true that most recent publications about rainfall thresholds for landslides have included some non-landslide inducing precipitation in developing or validating thresholds; it seems uncommon to continuously evaluate those thresholds against daily absence of landslides. This is an important point, because, if I understand correctly, you are arguing that the way that process-based landslide susceptibility models are being calibrated or validated (by comparison with past events) is biasing them in a way that decreases forecast accuracy. If so, and a new or different approach is needed for
validating models, what do you suggest?

REPLY: On the regional scale, landslides are very much binary events; they either happen (1) or not (0). This is a major difference to floods where continuous measurements (e.g. of streamflow) can be carried out. Continuous measurements for landslides are very much linked to the local scale where monitoring systems for progressively moving landslides are being used to measure e.g. displacement, soil moisture and rainfall. However, the results of such local scale monitoring systems can not easily be used for regional scale forecasts of landslides. Thus, landslides can be considered extreme events when they are analyzed on the regional scale. Understandably, model calibration with a limited number of extreme events is challenging and we argue that it is therefore questionable whether such forecasting models can be expected to be able to precisely predict future extreme events. Including rainfall events that did not trigger landslides is a useful strategy for validating model outputs; however, it is also not entirely straightforward as landslides often take place undetected or unreported. It is beyond the scope of the paper to propose new approaches for the calibration and validation of regional landslide forecasting models. It was rather our intention to draw intention to this issue and to provide basis for discussion.

Focusing on the absence of landslides as a criterion can be problematic at the regional scale, as landslides (e.g. in forests) may remain undetected. Page 18, line 27-29, The problem here is not just a matter of lowering model sensitivity. In many cases, available data are not adequate to create a very sensitive model using any procedure. For example, in the rolling hill terrain of the Esino River Basin, Gioia et al. (2015) found there was no well-defined relationship between topographic variables and landslides, making it difficult to attain high model sensitivity. For other areas, such as the Colorado Front Range (Alvioli and Baum, 2016), model sensitivity was impaired by the quality of available DEMs, such as those derived from legacy photogrammetrically mapped topographic contour data, as well as other data uncertainties. Mergili et al. (2014b) similarly seem to have experienced difficulties obtaining high sensitivity even when

NHESSD
using a 3-D method and ranges of depth, cohesion and friction. While I appreciate main the point of your paper that accounting for uncertainty through the use of model ensembles will improve forecasts, it seems clear that continued research is needed on other fronts as well to overcome some of the challenges in making accurate forecasts.

REPLY: Thank you for your comment. We will describe the problem (i.e. the inability to create a sensitive model due to data) in our revised manuscript.

Page 19, lines 8-9, What do you mean here by averaging performed by the infinite slope stability model? Don't 3-D models such as the model of Xie et al. (2003, 2004, 2006), r.slope.stability (Mergili et al. 2014a), and Scoops3d (Reid et al. 2015) perform a sort of averaging over the neighborhood of each point in the search grid? By considering a group of neighboring cells in each trial failure, the 3-D models effectively average out effects of the ground surface irregularities at the same time as they account for effects of the finite extent and lateral boundary effects of realistically shaped trial landslides.

REPLY: Yes, thank you for this comment. Indeed, this paragraph was formulated in a misleading way. In the revised manuscript we will rephrase it as follows: "The most commonly applied modeling approach relies on the infinite slope stability model which reduces the landslide geometry to a slope-parallel layer of infinite length and width. This leads to very pronounced patterns of the factor of safety, whereas modeling approaches that introduce more complex landslide geometries produce smoother results since the effects of neighboring pixels are averaged out. Whether complex approaches such as r.slope.stability (Mergili et al. 2014a), Scoops3d (Reid et al. 2015) or approaches based on slip circles or ellipsoids (Xie et al. (2003, 2004, 2006) are able to outperform the infinite slope stability model depends on the settings, notably the landslide geometries. In theory, the infinite slope stability model is suitable for shallow landslides with length-to-depth rations above 18-20 (Griffiths et al., 2011; Milledge et al., 2012)."

Griffiths, D. V., Huang, J., and de Wolfe, G. F.: Numerical and analytical observations
on long and infinite slopes, Int. J. Numer. Anal. Met., 35, 569–585, 2011 Milledge, D., Griffiths, V., Lane, S., and Warburton, J.: Limits on the validity of infinite length assumptions for modelling shallow landslides, Earth Surf. Proc. Land., 37, 1158–1166, 2012.

Technical corrections: Page 2, line 29, change "explicitly introduces in into the model" to "explicitly introduces it into the model" Page 6, line 24, capitalize "it" Page 8, line 22, change "provide" to "provides" Page 8, line 31, change "if" at the end of the line to "it" Page 13, line 13, change "a proof on concept" to "a proof of concept" Page 15, line 16, change "parameter" to "parameters" Page 16, line 22, delete comma after "data assimilation applications in both" Page 18, line 16, change "calibration if capable" to "calibration is capable" Page 18, line 27-28, change "had to be" to "must be" Page 18, line 30, change "a nonsuperior state over" to "an inferior state below" Figure 4, add outline of area shown in Figure 5.

REPLY: All suggested technical corrections will be integrated into the revised manuscript.

References cited: Alvioli, M. and Baum, R. L.: Parallelization of the TRIGRS model for rainfall-induced landslides using the message passing interface, Environmental Modelling & Software, 81, 122–135, doi:10.1016/j.envsoft.2016.04.002, 2016. Gioia, E., Speranza, G., Ferretti, M., Godt, J.W., Baum, R.L., Marincioni, F., 2015, Application of a process-based shallow landslide hazard model over a broad area in Central Italy: Landslides, doi: 10.1007/s10346-015-0670-6, p. 1-18. Greco, R. and Pagano, L.: Basic features of the predictive tools of early warning systems for water-related natural hazards: examples for shallow landslides, Natural Hazards and Earth System Sciences Discussions, 1–31, doi:10.5194/nhess-2017-269, 2017. Mergili, M., Marchesini, I., Alvioli, M., Metz, M., Schneider-Muntau, B., Rossi, M. and Guzzetti, F.: A strategy for GIS-based 3-D slope stability modelling over large areas, Geoscientific Model Development, 7(6), 2969–2982, doi:10.5194/gmd-7-2969-2014, 2014a. Mergili, M., Marchesini, I., Rossi, M., Guzzetti,
F. and Fellin, W.: Spatially distributed three-dimensional slope stability modelling in a raster GIS, Geomorphology, 206, 178–195, doi:10.1016/j.geomorph.2013.10.008, 2014b. Reid ME, Christian SB, Brien DL, Henderson ST (2015) Scoops3DâËŸAËĞ Tsoftware to analyze three-dimensional slope stability throughout a digital landscape. Virginia: U.S. Geological Survey Techniques and Methods 14-A1, 218 p., https://dx.doi.org/10.3133/tm14A1 WMO: Guidelines on Ensemble Prediction Systems and Forecasting, World Meteorological Organization, WMO-No. 1091, Geneva. Available at: http://www.wmo.int/pages/prog/www/Documents/1091 en.pdf, last access: 30 November 2017, 23 pp., 2012. Xie, M., Esaki, T., Zhou, G., and Mitani, Y.: Threedimensional stability evaluation of landslides and a sliding process simulation using a new geographic information systems component, Environ. Geol., 43, 503-512, 2003. Xie, M., Esaki, T., and Zhou, G.: GIS-based Probabilistic Mapping of Landslide Hazard Using a Three-Dimensional Deterministic Model, Nat. Hazards, 33, 265-282, 2004. Xie, M., Esaki, T., Qiu, C., and Wang, C.: Geographical information system-based computational implementation and application of spatial three-dimensional slope stability analysis, Comput. Geotech., 33, 260-274, 2006.

**NHESSD**

---

## Author Comment (AC2) · 15 May 2018

Dear referee #3, dear editor.

Thank you for your comments and suggestions on our manuscript. We have addressed your general and specific comments below.

Kind regards,

The authors

Anonymous Referee #3

General comment

[Figure]

The article "Probabilistic landslide ensemble prediction systems: Lessons to be learned from hydrology" presents an analysis of ensemble prediction systems in order to apply them in the probabilistic prediction of landslide occurrence. The paper is very long, especially in the introduction. However, it is well written, in a good English language. It follows somehow the IMRaD structure, even with some drawbacks, that should be improved. The subject is within the topic of special issue of NHESS journal. In my opinion, the manuscript needs major revisions before being accepted for publication. Mainly, the theoretical background described in the introduction is extremely long! I suggest a strong revision of this part aimed at shortening it. The same is for the conclusions section, which can be shortened. There are some parts of the introduction that should be moved in the discussion. On the other hand, the description of the method used for validating the background is quite fast, as for the results and discussion. Moreover, it seems that the Authors applied a method different from that extensively presented and discussed in the introduction. All these issues should be addressed in the revised version of the paper.

REPLY: Thank you for your remarks. We agree that our introduction and conclusion sections are lengthy. During our manuscript revision we intend to focus on shortening these sections. We will also critically review whether some parts can be moved to the discussion section. Moreover, we will aim to make it clearer how the methodology applied in our case study relates to the concepts and methods reviewed in the first part of the manuscript.

Specific comments

At page 1, lines 28-30, Authors state: "In this paper, we use prediction systems synonymously with early warning systems for terminological consistency within the landslide community although we acknowledge that early warning should also cover dissemination and response strategies". I strongly disagree with this terminological association. As acknowledged, an early warning system includes a prediction system and many other components. Thus, if the "landslide community" has used the two terms as synonyms since now, this paper could be a milestone in proposing a separation between them. I suggest to distinct the two items.

REPLY: Yes, we agree with this suggestion. In the revised version of the manuscript, the two terms will be distinguished.

Page 3, lines 23-24: Also, Intrieri et al. (2013) have presented a complete scheme for landslide early warning systems.

REPLY: Thank you for the suggestion; this reference will be included and we will highlight that Intrieri et al. (2013) provide a framework for the implementation of landslide early warning systems.

Page 4, lines 13-14: Please give some examples of prototypal landslide early warning systems.

REPLY: We will include a selection of prototypal landslide early warning systems with examples from New Zealand (Schmidt et al., 2008), Italy (Aleotti, 2004; Sirangelo and Braca, 2004), Japan (Sakai, 2008), Indonesia (Liao et al., 2010) and Germany (Bell et al., 2010; Thiebes et al., 2013).

Aleotti, P.: A warning system for rainfall-induced shallow failures, Eng. Geol., 73(3–4), 247–265, 2004.

Bell, R., Mayer, J., Pohl, J., Greiving, S. and Glade, T., Eds.: Integrative Frühwarnsysteme für Gravitative Massenbewegungen (ILEWS) - Monitoring, Modellierung, Implementierung, Klartext, Essen, Germany., 2010.

Liao, Z., Hong, Y., Wang, J., Fukuoka, H., Sassa, K., Karnawati, D. and Fathani, F.: Prototyping an experimental early warning system for rainfall-induced landslides in Indonesia using satellite remote sensing and geospatial datasets, Landslides, 7(3), 317–324, doi:10.1007/s10346-010-0219-7, 2010.

Sakai, H.: A warning system using chemical sensors and telecommunication technologies to protect railroad operation from landslide disaster, in Landslides and Engineered Slopes: From the Past to the Future, edited by Z. Chen, J.-M. Zhang, K. Ho, F.-Q. Wu, and Z.-K. Li, pp. 1277–1281, Taylor & Francis, Xi'an, China., 2008.

Schmidt, J., Turek, G., Clark, M. P., Uddstrom, M. and Dymond, J. R.: Probabilistic forecasting of shallow, rainfall-triggered landslides using real-time numerical weather predictions, Nat. Hazards Earth Syst. Sci., (8), 349–357, 2008.

Sirangelo, B. and Braca, G.: Identification of hazard conditions for mudflow occurrence by hydrological model:: Application of FLaIR model to Sarno warning system, Eng. Geol., 73(3–4), 267–276, 2004.

Thiebes, B., Bell, R., Glade, T., Jäger, S., Mayer, J., Anderson, M. and Holcombe, L.: Integration of a limit-equilibrium model into a landslide early warning system, Landslides, 11(5), 859–875, doi:10.1007/s10346-013-0416-2, 2013.

Page 7, lines 1-3: I think that the time between triggering/propagation and collapse stages varies also according to the landslide types.

REPLY: Yes, this will be mentioned accordingly in the revised manuscript.

Section 7.2.1 is a description of a model, thus it should not be in the "Case study" section.

REPLY: In the revised manuscript we will split the chapter: 7 will introduce the modelling approach, and chapter 8 will describe the case study.

Page 11, lines 18-19: Recently, Tran et al. (2017) proposed an application of TRIGRS with a 3D model to analyze 3D slope stability.

REPLY: Thanks a lot for this remark – we will include the reference to Tran et al. (2017) in the revised manuscript.

At page 12, line 7, Authors state that they computed 25 model runs. At page 13, line 20, they refer to 24 model iterations. Please explain.

REPLY: The figure only shows 24 model runs out of the 25 completed iterations. For the revised manuscript, we will modify the figure to only show 6 or 9 selected model runs (as suggested in your comments). We will also explain that these only represent a selection of model runs to visualize the different modeling outcomes.

Please check all the brackets in the text: somewhere, in particular in relation to references, there are many of them.

REPLY: We will check the brackets and delete the ones not necessary.

Please check the acronyms in the text. Use always acronyms after defining them.

REPLY: We will check all acronyms to make sure that acronyms are being used after they were introduced.

Figures

Figure 1. I suggest to add a map of the whole Austria with the indication of Lower Austria (for non-European readers).

REPLY: Yes, we agree. We will change the figure to also show an overview map of Austria.

I can't understand the meaning of the elevation classes. I suggest using a continuous scale.

REPLY: Thank you. We will omit the classified elevations and use a continuous scale as suggested.

Figure 3. This figure is extremely hard to see and read. It's quite impossible to see the differences between the different maps. I suggest to split it in 2 or 3 figures or to leave in the text just 6 or 9 significant cases and to put the other maps in an ancillary file.

REPLY: Thank you for this suggestion. We will follow this suggestion in our revision.

Figure 4. There are incongruities in the legends of "probability of failure" and "building

exposure". As an example, a pixel with probability of failure (or building exposure) equal to 0.25 is in the first or in the second class? The same for values equal to 0.50 and 0.75. Please correct them including or excluding the extremes in the classes as appropriate.

REPLY: Thank you for pointing this out. We will change the class breaks during the revision.

Why the forested areas are reported in the map? Figure 5.

REPLY: We will omit the forest area in the revised version of the figure. In addition, we would like to add a hillshade to the figure so that the reader gets an impression of the topography.

As for the previous figure, I suggest to correct the incongruities in the legends and to explain why the forested areas are reported in the map.

REPLY: As in our reply to your comments on figure 4, we will correct the legend of this figure and omit the forest in the revised version of the manuscript.

I can't understand the symbols used for indicating landslide head scarps. If two dimensions are not needed, I suggest using a point layer.

REPLY: We chose this symbol to increase the visibility of the landslide points. However, following your suggestion, we will use a simple point symbol in the revised figure.

TECHNICAL CORRECTIONS

Page 3, line 14: please delete "p. 1".

Page 3, line 16: please delete "p. 1".

Page 9, line 1: please correct "this event".

Page 10, line 16: please correct "km2".

Page 10, line 18: please change "," into ";" in the references.

Page 13, line 10: insert "of" in "representation surface topography".

Page 16, line 30: Replace "a couple" with "some".

Page 18, line 7: correct "landslides types".

REPLY: All technical corrections will be carried out as suggested.

REFERENCES

Intrieri, E., Gigli, G., Casagli, N., and Nadim, F.: Brief communication "Landslide Early Warning System: toolbox and general concepts". Nat. Hazards Earth Syst. Sci. 13, 85–90, doi:10.5194/nhess-13-85-2013, 2013.

Tran, T.V., Alvioli, M., Lee, G., and An, H.U.: Three-dimensional, time-dependent modeling of rainfall-induced landslides over a digital landscape: a case study. Landslides, doi:10.1007/s10346-017-0931-7, 2017.

---

## Author Comment (AC3) · 28 May 2018

Dear referee #2, dear editor.

Thank you for your extensive review! Please find our replies to the issues raised by you below.

Kind regards, The authors

Anonymous Referee #2

General comments Landslide early warning is a very interesting and relevant topic. However, in my view the authors fail to deliver what they promise. After reading a 10 page long introduction, I was convinced that the authors were going to introduce a

new framework/method to incorporate ensembles of weather forecasting for landslide prediction, similarly to what has been done for hydrological forecasting. However, that is not the case. The authors selected a landslide physically based model, which they run considering uncertainty in some pre-selected model parameters. Nowhere, did the authors considered uncertainty introduced by the weather forecast that is essential in early warning. While it is valid to use a Monte Carlo approach to evaluate uncertainty introduced by the parameters of the landslide model (many studies have done this, both in the hydrology and the landslide community), this is different from assessing uncertainty in a weather forecast (by using an ensemble for weather forecasting, provided by operational forecasting system such as those listed on page 6, lines 12-17) particularly relevant for early warning systems. So my main concern with this paper is that while the reader is given the impression that the paper is about uncertainty in weather forecasts to improve landslide early warning systems, the paper is about something else, i.e. uncertainty in landslide model parameterisation. Why then writing so much about early warning and ensembles for weather forecasting (one example is the entire section 4, but I will provide more examples in the Specific Comments section) if the paper is about something else? I believe that this study could be a really interesting if the authors have indeed used landslide physically based models together with many different initial conditions of rainfall (based on a possible weather forecast), to enhance a landslide early warning system at a regional scale. Given the results presented, in my opinion this paper needs to be restructured so that the first six sections reflect what the paper is really about, that is uncertainty related to the geotechnical parameterisation of the landslide model, as stated on page 10, line 5.

REPLY: We agree that a restructuring of the first sections of the manuscript is necessary to avoid the confusion that you have described. Moreover, we agree that the revised version of the manuscript should indicate clearly what it will cover. To better reflect the structure of the paper as well as the content we intend to change the last paragraph of section 1 to: "The overall aim of this paper is to form a basis for discussion on how probabilistic landslide forecasting and early warning systems can be

implemented. To this end, we provide a review on how probabilistic modeling methods and in particular ensemble predictions are applied for hydrological forecasts, and how these deal with uncertainties. Moreover, we highlight challenges and limitations for the calibration of models focusing on extreme events such as landslides. In a case study application for Austria, we present a simplified framework of a landslide ensemble forecasting system in which the geotechnical parameters are treated probabilistically. In addition, we present suggestions on how probabilistic landslide forecasts can be visualized in a way that stakeholders can base their decisions on. We conclude the paper by putting forward a selection of challenges that we hope will facilitate the discussion of the topic and will ultimately lead to increased efforts for probabilistic landslide forecasting. "

Other major comments include: -Clearly state what is novel about this paper and its direct relevance for early warning. Monte Carlo simulation as a way to incorporate parameter uncertainty is not new (just few examples in a landslide context, among many, Haneberg, 2004; Cho, 2007; Melchiorre and Frattini, 2012).

REPLY: Yes, we agree that the relevance and the novelty of our paper should be described more clearly. We intend to add the following paragraph: "With this paper, we want to identify a gap in the prevalent landslide forecasting methods. Ensemble predictions and the explicit integration of uncertainties in forecasts are widely used in the fields of meteorology and hydrology; however, such activities are not as common for landslide forecasting. We therefore think that the landslide community could benefit from the experiences of the neighboring disciplines and that our paper can provide a starting point for increased efforts into these directions. An important novelty of our paper consists in the presentation of a landslide forecasting framework utilizing the physically based landslide model TRIGRS which we implemented within an open source environment. With our case study, we highlight how ensemble prediction for landslides could be implemented as operational systems."

-Lack of justification on why only certain model parameters were considered uncertain

and not others.

REPLY: With our case study, we want to present how a physically based model can be utilized to provide landslide forecasts based on ensemble predictions. The system is thought as a proof of concept and not as an operational system. We think that the general scheme becomes clear although we only allow for uncertainty of selected parameters. Still, the developed system would technically be capable of ingesting uncertainties for all parameters, including weather forecasts. We will add this information to the revised manuscript.

-Not enough detail is provided on the probability distributions used for the uncertain parameters, which is essential for the reproduction of the results.

REPLY: We understand the point of the reviewer. We deliberately omitted detailed information on the probability distributions to avoid distracting the readers. However, in the revised manuscript we will add additional information on the geotechnical parameters and their probability distributions. Please also see our replies to the comments on these issues in the section on specific comments.

-The paper lacks a proper structure. It seems more like a thesis chapter, than a paper for a journal. The Introduction is too long (six separate sections), the discussion is incomplete, and the main conclusions that arise from this study are non-existent. What the authors call Conclusions section, reads more like a Discussion section. The fact that the study lacks a Method section, jumping straight from Introduction to Case Study, makes the paper more difficult to read.

REPLY: Yes, we agree that restructuring the paper during revision is necessary. The introduction section will be shortened, the case study will receive a proper method section (presently, the method description is mixed with the description of the study area), and the discussion and conclusion section will be revised.

Specific comments Page 1, line 13: What do the authors mean by "larger scales"?

REPLY: Apologies, the terms large and small scales have been mixed up in the manuscript. To avoid confusion, we will use the terms such as small and large regions in the revised manuscript.

Page 1, line 14: Why convective-scale numerical weather predictions specifically and not other numerical weather predictions? Please also briefly explain what convective-scale numerical predictions are.

REPLY: In our paper, we distinguish three types of numerical weather predictions: global, regional and convective-scale (page 6 line 3). The latter provide the highest spatial resolution and can be considered the most useful for forecasting landslides on the regional scale.

Page 1, lines 17-20: Is this study going to answer any of the future research directions identified by the authors in the Abstract? Or are the future research directions identified here, gaps that need to be addressed in the future? If the first, please make clear in the Abstract what the main conclusions of this study are and how this study contributes to the points raised. If the latter, before pointing the future directions of research, please make clear what the main conclusions and contributions of this study are.

REPLY: The abstract will be rewritten to better reflect the content of our paper and the part on future research directions will be omitted.

Page 1, line 31 – Page 2, lines 1-2: The authors say that "For natural hazard types with a rapid onset (. . .) rainfall can be considered as the main triggering mechanism." What about earthquake triggered landslides? They also have a rapid onset and are not triggered by rainfall. Or wildfires, etc.?

REPLY: Yes, you are of course right; we chose the wording poorly. We will change the sentence to: "Rainfall triggered natural hazards with a rapid onset such as landslides and flash floods greatly benefit from rainfall nowcasting or short-term rainfall forecasting."

Page 2, lines 7-12: The authors say on page 2 that there are multiple reasons why numerical weather predictions are not used within the landslide early warning community. One of those reasons pointed by the authors is the "the complexity of single landslide detachments: the same landslide triggering event does not necessarily cause other landslides as the time between propagation stage and the collapse phase may vary significantly based on differences in local conditions (. . .) and spans from minutes (. . .) to years (. . .)". While it is true that the same rainfall event may result in a landslide in certain places (and for certain initial conditions) but not in others, and that the collapse phase may vary from minutes to years, I do not fully understand how this relates to the lack of uptake of numerical weather predictions by the landslide early warning community. Please make the link clearer to the reader.

REPLY: We apologize for the poor wording. During the manuscript revision this section will be rewritten. In fact, we think that the complexity of landslide forecasting is an important reason to the relatively low number of operational landslide early warning systems.

Page 2, lines 25-29: The authors claim that "This paper reviews and summarizes concepts of ensemble prediction systems (EPS) in hydrology and how those can be translated to be applicable also in process-based landslide early warning systems. A strong emphasis is put on how to deal with spatial uncertainties by demonstrating the benefits of probabilistic model application which does not eliminate uncertainty, but it explicitly introduces in into the model results." While the authors provide a case study where they consider spatial uncertainties and how these uncertainties impact the model output (Factor of Safety), I do not agree that the authors show how ensemble predictions systems can be used in process-based landslide early warning systems. For early warning systems, it is essential to consider the uncertainty in weather forecasts, and not just the uncertainties introduced by model input parameters as the authors do in their case study. Please make the text here more accurate, so that it reflects what is later delivered.

REPLY: Thank you for this comment. First of all, we will change the term early warning system to forecasting system. And we agree that an operational landslide forecasting and / or early warning system should also include the uncertainties of weather forecasts. However, our study is thought as a proof of concept and not as an example of an operational landslide forecasting system. As a consequence, we think it is justified to only allow for uncertainty of all model parameters. But of course, from a technical point of view the described framework is also be able to include a selection of differing weather forecasts.

Page 2, lines 31-33, Page 3, lines 1-2: The authors state at the end of Section 1, that the aims of the paper are: "a) to critically evaluate the current state of physically based landslide early warning, its limitations and possible ties to hydrological forecasting; b) on this basis, to foster cooperation across disciplinary boundaries to bring together scientists from different fields to pursue research based on forecasting experiences gained in the last couple of years". Is this paper a literature review on this topic, or does it aim to introduce something new? If the first, please make it clear. If the latter, please clarify what are the research questions that the authors aim to answer.

REPLY: As described in our replies to your general comment, the last part of the introduction section will be rewritten to better reflect the content and structure of the paper, as well as the relevance and novelty. The section you commented on here will be omitted.

Page 3, lines 29-30: Streamflow measurements for extremely high conditions are difficult to obtain and far from accurate. Please consider rephrasing the sentence in your manuscript. REPLY: We agree. We will rephrase this sentence to: "Despite considerable measurement uncertainties in phases of high flow, the prediction domain in flooding, which is usually streamflow, is more straightforward . . ."

Page 4, lines 10-11: How does this sentence link to what comes before? Why do the authors jump to data collection?

REPLY: The sentence will be omitted during revision.

Page 4, lines 12-25: Why do the authors talk about underestimation of landslide losses here? Why is this relevant for the overall argument that the authors making? Please clarify.

REPLY: We mention landslide losses here to underline that their damage potential is underestimated, underreported and often perceived as private losses. As a consequence, landslide forecasting and early warning systems are not as common as e.g. flood forecasting systems. In the revised manuscript, we this paragraph will be rephrased and moved to a more appropriate position.

Page 4, lines 27-28: Perturbations to model parameters is one thing, perturbation to starting/initial conditions is another (as it is clear from the definition given by WMO that the authors quote in page 3, lines 5-11). It is crucial to clarify what this study is about. Up to this point, the reader is made to believe that perturbed initial conditions (reflecting uncertainty in weather forecasts) are going to be used (and maybe also different combinations of physical parameterisation schemes of the landslide model). However, that is not the case. The study is about uncertainty in the physical parameterisation of the landslide model. Please note, that I acknowledge that the authors make a more clear distinction later on page 5, lines 5-9, when they define the term "ensemble prediction" for multi-parameter and multi-model predictions. But my point is that, the reader does not know what the paper is about. And if the paper is going to be about parameter uncertainty (not clear at this point), why do the authors spend so much time talking about forecasting, weather ensemble forecasting and early warning?

REPLY: Yes, we agree that the present manuscript may confuse the reader about the content of the paper making them believe that we will include uncertainty of weather predictions (which we don't). As mentioned earlier, the revised manuscript, we will describe more clearly what the content is. Moreover, during the requested shortening of the theoretical / review part we will include less information on weather ensemble

predictions.

Page 5, line 10: I do not agree that there have been only a few attempts to use ensemble techniques in landslide research (assuming here that the authors are talking about multi-parameter and multi-model ensembles). Some examples include: Haneberg, 2004; Rubio et al, 2004; Cho, 2007; Melchiorre and Frattini, 2012, Arnone et al, 2014. Some of the references provided by the authors later (page 5, line 27) are possibly also good examples.

REPLY: Yes, we agree. We will appropriately acknowledge the work suggested by the reviewer in the revised manuscript, and rephrase the text accordingly.

Page 5, line 12: Once again when the authors say that "None of them, however, incorporate ensemble techniques in real-time application", makes me believe that this study will address that, what in my opinion is not the case.

REPLY: Yes, we do not present an operational forecasting system which provides forecasts in real-time. However, we present a framework that is from a technical point of view able to deliver such information. With the respective sentence, we wanted to highlight the fact that at present, there are no real-time landslide forecasting systems in operation that are based on ensemble prediction techniques. We think that this information is important and should be mentioned here. However, we did not want to claim that our study provides such as system. We think that the changes related to the description of our study's content (see comments above) will make it clear that no operational system is presented in our paper.

Page 5, lines 28-32: Similarly to the previous comment, when the authors say that "Haneberg (2004), Park et al. (2013), Raia et al. (2014), Lee and Park (2016) and Zhang et al.(2016) treat soil properties at regional scale applications in a probabilistic way by randomly selecting variables from a given probability density function, mostly by means of Monte Carlo (MC) simulation (. . .) None of those probabilistic approaches are operated in spatial real-time-early warning systems, not even on a prototype basis",

makes me believe that this study will go beyond using Monte Carlo simulations of soil properties randomly sampled, and provide a method/case study/framework to be used it in real time warning systems. That does not turn out to be the case.

REPLY: Yes, we agree that section might confuse readers. As stated in a previous reply, with our revision we will make clear what our study is about. In relation to this paragraph, we would describe our study as a presentation of a technical landslide forecasting framework in which some parameters values (i.e. soil properties) are treated probabilistically (i.e. randomly chosen from a predefined parameter range) and which could be operated in a real-time manner

Page 6, line 28: What do the authors mean by hydrological applications?

REPLY: We refer to flood forecasting, and will write that in the revised manuscript.

Page 7, lines 8-26: Again, why do the authors spend so much time talking about quantitative precipitation forecasts (QPF) from numerical weather predictions, including giving the example of flash floods that require QPFs with 1-6 hour lead times, if the study does not use such QPFs?

REPLY: Yes, we do not use QPF in our study because we aim for a proof of concept of our prototypal technical landslide forecasting system. We will shorten this section during the revision and omit unnecessary details on flash flood forecasting.

Page 8, lines 5-6: This is a confusing sentence mixing equifinality and nonlinearity. Please consider rephrasing it.

REPLY: Thank you. Yes, we will rephrase this sentence.

Page 8, line 31: The concept Factor of safety is introduced here (page 8, line 31), and defined later on page 11, lines 21-24.

REPLY: We are sorry about the repetition. We will only explain the FoS concept once in the revised manuscript.

Page 9, lines 11-13: The sentence "Commonly, calibration will improve the reliability of forecasts (i.e. the match of the target variable or forecast probabilities to frequency of observations of the event) but reduce the resolution of the forecast (the ability to discriminate whether an event will occur or not)." appears twice in this manuscript (page 9, lines 11-13, and page 18, lines 17-19). Please do not repeat the exact same text twice in the same manuscript. It is also unclear what the authors exactly mean. I have not seen before "resolution of forecast" being defined as "the ability to discriminate whether an event will occur or not"

REPLY: We are sorry about the repetition. We will rephrase this part and only mention it once in the revised manuscript.

Page 9, lines 13-14: "calibration will improve forecasts of common events, but will also lead to the underprediction of more extreme events."- Please add reference(s).

REPLY: We will add WMO (2012) as a reference. WMO: Guidelines on Ensemble Prediction Systems and Forecasting, World Meteorological Organization, WMO-No. 1091, Geneva. Available at: http://www.wmo.int/pages/prog/www/Documents/1091_en.pdf, last access: 30 November 2017, 20 23 pp., 2012.

Page 9, lines 17-26: This paragraph is confusing. At the beginning of the paragraph the reader is given the impression that the authors are going to talk about validation. But then the paragraph seems to be about calibration.

REPLY: Yes, we agree and will revise and restructure the paragraph to distinguish more clearly between calibration and validation.

**Page 9, lines 24-25: What do the authors mean by "dependence on the model structure"?**

REPLY: We meant the general laws and function of the used model. We will omit this in the revised manuscript as it is not necessary to mention this.

Page 9, line 33 and page 10, lines 1-2: What is the point of providing these two refer-

ences here, without a brief description of the measures? If those measures are relevant for the case study, please consider giving a brief description here. If not, I am not sure why the authors introduce the references here.

REPLY: We will omit this in the revised manuscript.

Page 10, line 3: After 10 pages of introduction (not sure why it has to be so long!), it would be helpful to state clearly what are the challenges that are going to be addressed in this paper and the research questions, before moving to the case study section. So far, this reads more like a book/thesis than a journal paper. Jumping from introduction to case study is also not very common.

REPLY: Yes we agree with all these points. During the manuscript revision, the introduction section will be shortened substantially; the addressed challenges will be clearly stated; and the following section will be restructured.

Page 10, line 4: What do the authors mean by "simplified ensemble modelling"? What has been simplified?

REPLY: Yes, this should be made clearer in the revised manuscript. Simplification here refers to the limited number of parameters which are dealt with in a probabilistic way.

Page 10, lines 5-7: It is unclear what is done in this study regarding point c) "how infrastructure data can further supplement early warning procedures in an exposure context." Later on in the study (page 14, lines 3-5), the authors overlap the results of the landslide model for a past rainfall event with Open Street Map. However, given that this is based on a past rainfall event (and not forecasting weather) how can this be used to supplement early warning procedures? Please rephrase point c) to reflect the results shown later on, or consider deleting point c) from this list.

REPLY: Yes, we agree and will restructure and revise this section accordingly. In response to your comment on how using a past rainfall even can contribute to early warning, we would like to note that we want to present a technical landslide forecasting

prototype; this is why we consider using a past rainfall event as sufficient (as proof of concept). We agree that this was not made clear in our present manuscript.

In here the authors state more clearly what the aims of this case study are. Taking into account that uncertainty in weather forecasts in landslide prediction is not part of those aims, why spending so much text up to here on that that topic?

REPLY: Yes, the present version of the manuscript does not make it clear what the aim of the study was. As described in a previous reply, this will be clearly stated in the revised manuscript.

Page 10, lines 26-27: Please add references after "physically based models can be quite commonly found to evaluate rainfall-induced landslide susceptibility at the regional scale"

REPLY: We intend to add the following references: Ciurleo, M., Cascini, L. and Calvello, M.: A comparison of statistical and deterministic methods for shallow landslide susceptibility zoning in clayey soils, Eng. Geol., 223, 71–81, doi:10.1016/j.enggeo.2017.04.023, 2017. de Lima Neves Seefelder, C., Koide, S., Mergili, M.: Does parameterization influence the performance of slope stability model results? A case study in Rio de Janeiro, Brazil. Landslides, 14(4), 1389–1401, 2017. doi:10.1007/s10346-016-0783-6 Park, H.-J., Jang, J.-Y. and Lee, J.-H.: Physically Based Susceptibility Assessment of Rainfall-Induced Shallow Landslides Using a Fuzzy Point Estimate Method, Remote Sens., 9(5), 487, doi:10.3390/rs9050487, 2017. Thiebes, B., Bai, S., Xi, Y., Glade, T. and Bell, R.: Combining landslide susceptibility maps and rainfall thresholds using a matrix approach, 17, 2017.

Page 12, lines 4-7: The description of the model setup is too generic. For example, what is the time step used? How many parameters does the model have?

REPLY: We will add more information on the model setup in the revised manuscript.

Page 12, lines 4-7: What are the properties of the normal distribution that the authors

sample from (i.e. mean and standard deviation) and how were they derived? On what grounds did the authors select a normal distribution and not any other distribution?

REPLY: This is the first of several comments on the geotechnical parameters and their probability distributions [the other comments are for (1) page 12 line 21; (2) page 12 line 25; (3) page 12 line 31-32]. We think that it is useful and more straightforward to reply to them in one lumped answer. For the following remarks on this issue, please refer to the following reply. We selected a normal distribution over another distribution based on findings of Wang et al. (2015) who concluded that lumping data from many different sources (i.e. different in situ soil sampling sites in this case) tends to result in a normal or lognormal distribution. This observation was also made on the extensive dataset of Tofani et al. (2017) whose data also matched such distributions. Since there is no way of establishing a perfect parameter distribution for such large areas, we are convinced that using a normal distribution is sufficient for now. In the present manuscript, we deliberately omitted mean values and standard deviations of the parameter distributions because we found no benefit in such singular parameters for our purpose, since we use uniquely sampled data out of the entire distributions that are based on a set max and min value for each model iteration. However, we intend to list mean values and standard deviations in the revised manuscript. The utilized max and min values were based not on measurements (as mentioned in the manuscript), but on a compilation of multiple geotechnical textbooks that publish 'typical' parameter ranges for the appropriate subsurface conditions (e.g.

Richwien and Lesny 2004, Smoltczyk 2001, Türke 1999). This will also be explicitly mentioned in the revised manuscript. Richwien, W., Lesny, K., 2004. Bodenmechanisches Praktikum, Auswahl und Anwendung von bodenmechanischen Laborversuchen. 11 ed., Verlag Gluckauf GmbH, Essen. Smoltczyk, U., 2001. Grundbau-Taschenbuch. Teil 1: Geotechnische Grundlagen. 6 ed., Ernst & Sohn, Berlin. Turke, H., 1999. Statik im Erdbau. 3 ed., Ernst & Sohn, Berlin.

Page 12, lines 7: Why 25 model runs? Such a small number may not adequately

represent the parameter space.

REPLY: Yes, we agree that more model runs would be able to better represent the entire parameter space. But for a proof of concept we think that 25 model runs are appropriate. Moreover, without the discussed parallel computation model iterations were relatively time-demanding.

Page 12, line 8: What is the "initial model run"? Is it the first of the 25 model runs?

REPLY: Yes. We will rephrase this to make it clearer in the revised manuscript.

Page 12, lines 9-12: Please consider rephrasing this sentence, as it is confusing. The authors say that the probability of failure of a given cell is equal to dividing the number of unstable raster cells by the number of model runs. It is confusing as for "this raster cell" they count the "number of unstable raster cells". I suppose that the authors mean the number of simulations that lead to FoS<1 for that specific raster cell, divided by the total number of simulations (i.e. 25). Is that the case? If so, please change the text to reflect that.

REPLY: Yes, we were indeed referring to the number of simulations that lead to a FoS<1. We will rephrase the sentence accordingly.

Page 12, line 21: Is the data from Tofani et al. (2017) mentioned here used by the authors to derive the parameters of the probability distributions of the uncertain parameters (i.e. mean and standard deviation of the normal distributions)? If so, and as mentioned in a previous comment, please provide the values of mean and standard deviation of the distributions and how those values were determined.

REPLY: No, we used published values from multiple geotechnical textbooks that describe 'typical' parameter ranges for the appropriate subsurface conditions (e.g. Richwien and Lesny 2004, Smoltczyk 2001, Türke 1999). This will also be explicitly mentioned in the revised manuscript. Please also note our reply to the comment on page 12, lines 4-7.

Page 12, lines 25-26: "(. . .) the boxplots suggested normal to lognormal parameter distributions. This is a common observation and might be a result of the central limit theorem". A lognormal distribution is not the result of the central limit theorem.

REPLY: We agree. This part of the sentence will be omitted.

Page 12, line 25: If the boxplots suggest normal to lognormal distributions, why did the authors decide on normal distributions? Please justify the choice made. Is the observation "the boxplots suggested normal to lognormal parameter distributions" valid for all model parameters? Please detail which parameters show a normal distribution and which parameters show a lognormal distribution.

REPLY: Please refer to our reply on your remark on page 12, lines 4-7.

Page 12, line 30: Why do the authors introduce GLUE here? GLUE indeed involves Monte Carlo simulation, but there is more to GLUE than that. In this study, Monte Carlo simulations are performed, but GLUE is not. So I do not see the need to mention GLUE here. It may only contribute to confuse readers that are not familiar with GLUE. REPLY: We agree with you and will not mention GLUE in the revised manuscript.

Page 12, lines 31-32: Are the parameters sampled from a predefined parameter range or are they sampled from normal distributions? Please clarify.

REPLY: Please refer to our reply on your remark on page 12, lines 4-7.

Page 12, lines 1-2 and line 33: Please justify why only uncertainties related to soil depth, effective cohesion and effective friction angle were considered. How do the authors know what are the most influential model parameters, without having carried out a sensitivity analysis?

REPLY: We considered this simplified approach (i.e. limited number of uncertain parameters) appropriate for our proof of concept. We will clearly note this in the revised manuscript.
Page 13, line 6 (caption of figure 2): What is "state" function?

REPLY: In the revised manuscript, we will omit the term "state" and rephrase the sentence to: "This gives us confidence to use plausible parameter ranges with a normally distributed function based on geotechnical textbooks to characterize soils in our study area."

Figure 2 (page 27): As mentioned in a previous comment, please provide the parameters of the normal distributions. Some of the points in Figure 2 do not seem to come from a normal distribution (e.g. for friction angle the distance between the 25th percentile and 50th percentile is quite different from the distance between the 50th percentile and 75th percentile).

REPLY: In the revised manuscript, we will list mean values and standard deviations for our geotechnical parameter distributions. Please also see our reply to the comment on page 12, lines 4-7

Page 12, line 7 and page 13, line 11: Is the model run for a specific rainfall event? Why 3 hourly time steps? Was 3 hours the duration of the rainfall event? Please provide enough detail on the rainfall event used to run the landslide model and why that specific event has been selected.

REPLY: Yes, we used a 3 hour rainfall event for our study. We will add more information on the rainfall event.

Page 13, lines 13-14 and lines 20-21: This is where the study falls short, i.e. show how numerical weather predictions and related uncertainty could be incorporated in a real-time application. If the landslide model is run with rainfall over the last three hours (page 13, lines 20-21), instead of a rainfall forecast, this cannot be used for early warning. But wasn't early warning the main selling point of this study?

REPLY: Yes, our present manuscript does not make it clear that our study does not aim to provide actual real time forecast and early warnings but aims to present a prototype

of a technical landslide forecasting system in which parameter uncertainty is integrated. This will be changed during revision.

Page 13, line 20: Are there 24 or 25 model simulations?

REPLY: We calculated 25 model iterations but only present 24 model runs here for the purpose of visualization.

Figure 3 (page 28) and page 13, lines 23-25: The 24 figures are not very helpful, as the reader cannot see the differences between the maps. What is the message that figure 3 is trying to illustrate? Is it essential to show these 24 maps, given that the reader cannot see much?

REPLY: Yes, we agree. The figure will be modified to only show 6 or 9 model runs in the revised manuscript (as also suggested by another referee).

Based on the figures provided I cannot see whether "the results indicate quite significant changes across individual members, but also quite high similarities although parameters change drastically between some of the members" (page 13, lines 23-25)

REPLY: There are indeed substantial differences, however, there are hardly visible as the maps are so small. By only showing 6 or 9 selected maps, this will become more apparent.

Page 13, line 32: Please make clear what uncertainties are accounted for, given that only certain model parameters were considered uncertain, and other uncertainties such as model structural uncertainty and rainfall uncertainty, were not considered.

REPLY: Yes, we want to do that in our revised manuscript.

Page 14, line 2: What do the authors mean by "this specific time"? Please clarify.

REPLY: We agree that this is confusing. The part will be omitted.

Page 14, lines 7-8 and figure 4 (page 29): The results shown in this study refer to a

specific rainfall event. This should be made clear, namely in the main text and in the captions of figures 4 and 5. If it does not rain, the values shown in figures 4 and 5 are no longer the probability of failure. And if it rains a lot (meaning much more than the rainfall event used to run the landslide model) the probability of failure is much higher than what is shown in the figures. Please make clear what the probability of failure refers to.

REPLY: We agree and will follow your suggestion during revision.

Page 14, line 11: It is incorrect to say that a narrow ensemble spread is an expression of equifinality.

REPLY: We agree; a narrow ensemble spread is not necessarily an expression of equifinality. We will rephrase this sentence.

Page 14, lines 12-13: There is not enough evidence in the results shown in this manuscript to make such a statement, as the authors did not run the model for different rainfall events. How can the authors know that for a different rainfall forcing the location of possible slope failures is not different?

REPLY: Yes, we agree. We will rephrase the sentence.

Page 14, lines 28-29: Indeed, but there are equally also many landslide initiation points that do not correspond to high failure probability. Of course, this may have to do with the rainfall input used. But that is my point - the probabilities shown in the map only have meaning for the specific rainfall event used, which the reader does not even know what it was.

REPLY: We agree, there are several landslides that are outside of areas modeled as likely to fail. This is related to the fact that it is not known under which triggering rainfall conditions these landslide failed. And indeed, our calculated probability to failure zonation only refers to the specific rainfall event used in our study. As written in a previous reply, more information on the rainfall event will be added.

Page 14, line 32: "there are still many accounted uncertainties": Please list the unaccounted uncertainties (or at least some of them). REPLY: We discuss the unaccounted under the conclusion section (e.g. the challenge of dealing with parameter uncertainties at regional scale modeling, constraints introduced by the modeling approach itself, the challenges of how to deal with rare events in model calibration). However, we agree that this should be mentioned already beforehand; thus we will add more information on uncertainties at this point of the manuscript.

Page 15, line 6: Why are some real landslides missed?

REPLY: In our study, we "missed" landslides because there is no information available under which rainfall conditions the slopes failed, thus making a validation extremely difficult (also see discussion on page 14)

Page 15, lines 14-15: What do the authors exactly mean by "spatial confidence buffer"? Are these the coloured areas in figures 4 and 5? How does the "spatial confidence buffer" show a narrow ensemble of spread? Spread of what? How does that relate to an equifinal result? How does that relate to slope angle?

REPLY: Due to the fact that the vast majority of simulations identified similar areas as likely for failure, we can have some confidence in that prediction. At the same time, this could also be a manifestation of equifinality; or relatively low sensitivity to the geotechnical parameters (and at the same time high sensitivity to slope angle). We will rephrase the sentence and discuss this in more detail.

Page 15, lines 14-14: How do the authors know that slope angle is the main predetermining factor based on their results? No sensitivity analysis has been performed, to make such a statement. If this statement is based on other studies, that needs to be made clear. It is important to highlight that the most influential parameter highly depends on the probability distributions used.

REPLY: Please note our previous reply. And we agree that probability distributions can

have a dominant influence on model outcomes.

Page 15, lines 17-18: The statement "no matter what the geotechnical or hydraulic input parameters are, it will be always the same slope segments that will result the highest slope failure probability" is not accurate. Geotechnical and hydraulic input parameters may still matter, even if slope angle may have the greatest impact on the model results. A slope with a certain angle (and assuming the same rainfall event), may fail for a certain combination of geotechnical and hydraulic parameters and not fail for another combination of geotechnical and hydraulic parameters.

REPLY: Yes, we agree and we will rephrase the sentence accordingly.

Page 15, lines 18-19: It is unclear what the sentence "Slope failure probability will ultimately vary only based on the dynamic component (here: rainfall) or if a spatially distributed soil depth map is provided" mean. Does failure of probability only depend on rainfall and soil depth map? What about the other aspects, such as slope angle, friction angle, cohesion etc?

REPLY: Of course, the spatial variability strongly depends on the slope and on the geotechnical parameters. This is not well explained in this sentence: very often, a reasonable spatial variation of geotechnical parameters is not possible due to very fine-scaled patterns and missing data. The same is often true for soil depth. We will change the sentence to "Slope failure probability will ultimately vary only based on slope (spatially) and on the dynamic component, here rainfall (temporally), unless spatially distributed maps of the geotechnical and geohydraulic parameters and/or of soil depth are available".

Page 15, lines 26-27: How does that add large errors? Please expand. How does that compare to the approach introduced in this paper, where only a pre-selected rain- fall event has been used?

REPLY: In most cases, the exact triggering conditions of landslides are unknown but

are approximated, e.g. by using measurements from the closest rain gauge (which might be miles away behind some hills) or rainfall radar measurements – this is what we referred to when we noted that reported landslides and their triggering conditions can contain large errors.

Page 16, lines 13-14: What does the sentence "However, narrowing down uncertainties is a good first step, but not the be-all and end-all of ensemble approaches." mean? What does it mean "be-all and end-all of ensemble approaches"? How do the authors suggest narrowing down uncertainties?

REPLY: We meant this is the sense of the only or ultimate solution. As stated in the following sentences, we suggest investigating the reasons for differences in model predictions in order to entangle the complex relation of interacting parameters.

Page 16, lines 23-25: But the authors do not do use physically based predictions with blends of most recent quantitative precipitation estimates either. The authors fail to show how the approach/results they present in this study can be used for early warning. For ex- ample, could the model be run fast enough to be used for early warning based on the weather predictions? This is just a very simple example.

REPLY: Yes, you are right, we do not use actual rainfall forecasts. Instead, we present a system which from technically can integrate any kind of fore- or hind-casted rainfall as long it is in a grid file. Please note that the computational burden and simulation times are discussed later in the paper (page 19).

Pages 16 and 20, Conclusions section: Line 30: The Conclusions section must start by clearly stating what was learnt from this study, i.e. what the actual conclusions of this study are. Only after this, the authors should discuss any challenges and/or future direction.

REPLY: We agree and will do so in the revised manuscript.

Large parts of the Conclusions section should be moved into the Discussion section.

REPLY: We agree with your comment (which was also brought up by the other referees). We will restructure the discussion and conclusion sections.

Page 17, line 8: Please provide references after "(. . .) probabilistic treatment of input parameters for regional model application has seen a rise only in the last couple of years."

REPLY: We will use the following references in the revised manuscript. Mergili, M., Marchesini, I., Alvioli, M., Metz, M., Schneider-Muntau, B., Rossi, M. and Guzzetti, F.: A strategy for GIS-based 3-D slope stability modelling over large areas, Geoscientific Model Development, 7(6), 2969–2982, doi:10.5194/gmd-7-2969-2014, 2014a. Neves Seefelder, C., Koide, S. and Mergili, M.: Does parameterization influence the performance of slope stability model results? A case study in Rio de Janeiro, Brazil, Landslides, doi:10.1007/s10346-016-0783-6, 2016. Raia, S., Alvioli, M., Rossi, M., Baum, R. L., Godt, J. W. and Guzzetti, F.: Improving predictive power of physically based rainfall-induced shallow landslide models: a probabilistic approach, Geoscientific Model Development, 7(2), 495–514, doi:10.5194/gmd-7-495-2014, 2014.

Page 18, lines 2-3: "when using literature values instead": It is unclear what the authors mean. Are the authors saying that literature values should be used instead, and that sample measures should be discarded? Please clarify.

REPLY: We wanted to express that when working on large areas it is generally not purposeful to spend many resources on in situ measurements for model parametrization. We will rephrase the sentence in the revised manuscript.

Page 19, lines 15-17: I am not sure that this sentence makes sense. Furthermore, the results presented in this study do not allow the authors to make such a strong statement. I would expect that if we are looking into a smaller area, the variability of certain soil properties (or more generally input parameters) is smaller than if we would be measuring the same properties across a larger area. But most importantly, the authors do not show results to back up their statement.

REPLY: Yes, we agree that this statement is not well worded and we will rephrase it in a less harsh way. Here, we wanted to state that uncertainties always remain even with more measurements. Even for small areas, these uncertainties are huge and not necessarily smaller than for large areas. For example assessing soil properties once for every square meter of a study area would completely neglect all conditions within these areas and important parameter differences would be missed. We made this argument not only from a purely practical point of view but also to underline the philosophical implication; one should not believe that uncertainties disappear (or even decrease) with more data and more measurements.

Page 19, line 20: I do not follow the argument. According to the authors, is computation problem still a problem nowadays or not?

REPLY: Sorry, this is a translation error. It should be "as soon as computational power became available".

Page 20, lines 5-6: People have been using HPC in landslide modelling – please check the literature.

REPLY: We agree that there are some studies involving landslide modeling and HPC, however, they do not aim for landslide forecasting. In the revised version of the paper, we will rephrase the sentence to the following: "While HPC applications are common in meteorological (Bauer et al., 2015) and hydrological forecasting (Shi et al., 2015), there are only few landslide related studies (e.g. Mulligan and Take, 2017; Shute et al., 2017; Song et al., 2017), however, none aiming specifically at landslide forecasting."

Mulligan, R. and Take, A.: Momentum transfer during landslide tsunami wave generation, vol. 19, p. 11065. [online] Available from: http://adsabs.harvard.edu/abs/2017EGUGA..1911065M (Accessed 25 May 2018), 2017. Shute, J., Carriere, L., Duffy, D., Hoy, E., Peters, J., Shen, Y. and Kirschbaum, D.: The Benefits and Complexities of Operating Geographic Information Systems (GIS) in a High Performance Computing (HPC) Environment, AGU Fall Meet. Abstr.,

31 [online] Available from: http://adsabs.harvard.edu/abs/2017AGUFMIN31B0072S (Accessed 25 May 2018), 2017. Song, Y., Huang, D. and Zeng, B.: GPU-based parallel computation for discontinuous deformation analysis (DDA) method and its application to modelling earthquake-induced landslide, Comput. Geotech., 86, 80–94, doi:10.1016/j.compgeo.2017.01.001, 2017.

Technical corrections Page 1, line 16 – "how ties to. . ." – Please rephrase. Page 2, line 19: "Another reason for the negligence of physically based forecasting initiatives (. . .)" – Please rephrase Page 2, lines 24-25: "The hydrological community has recently adopted to those ad- vancements (. . .)" – Please rephrase. Page 2, line 29: "in into" – Please correct. Page 3, line 26: In the sentence "One reason why landslide forecasting is seemingly more challenging can be (. . .)", please state what landslide forecasting is more chal- lenging compared to, i.e. "One reason why landslide forecasting is seemingly more challenging THAN X can be (..)" Page 6, line 13: Delete the comma after "In". Page 7, line 16: "longer lead times (. . .)" Longer relative to what? Page 8, line 31: "(. . .) when if" – please correct. Page 9, lines 17-18: What do the authors mean by "either-or-situations"? Page 14, line 30: What situation are the authors referring to? Please clarify. Page 15, line 2: Please clarify what "This" refers to, or in other others explain what is quite detrimental. Is it explicitly accounting for uncertainty?

REPLY: All suggested technical corrections will be integrated into the revised manuscript.

References Arnone, E., Dialynas, Y. G., Noto, L. V. and Bras, R. L. (2014). Parameter uncertainty in shallow rainfall-triggered landslide modeling at basin scale: A probabilistic approach. Procedia Earth and Planetary Science, 9, 101-111, doi: 10.1016/j.proeps.2014.06.003. Cho, S. E. (2007). Effects of spatial variability of soil properties on slope stability, Engineering Geology, 92(3-4), 97-109, doi: 10.1016/j.enggeo.2007.03.006. Haneberg, W. C. (2004). A rational probabilistic method for spatially distributed land- slide hazard assessment, Environ-

mental & Engineering Geoscience, 10(1), 27-43, doi: 10.2113/10.1.27 Melchiorre, C. and Frattini, P. (2012). Modelling probability of rainfall induced shallow landslides in a changing climate, Otta, Central Norway, Climatic Change, 113, 413–436, doi:10.1007/s10584-011-0325-0. Rubio, E., Hall, J. W. and Anderson, M. G. (2004). Uncertainty analysis in a slope hydrology and stability model using probabilistic and imprecise information. Computers and Geotechnics, 31(7), 529-536. doi: 10.1016/j.compgeo.2004.09.002 Interactive comment on Nat. Hazards Earth Syst. Sci. Discuss., https://doi.org/10.5194/nhess- 2017-427, 2017.

---

## Author Response (AR1)

**Author response**

Dear editor, dear reviewers,

we are happy to inform you that we have finalised the revision of our manuscript.

As requested by the editor and the reviewers, we have carried out a major revision of the manuscript that was explicitly guided by the comments provided. In addition to the revised manuscript, we have uploaded a version with tracked changes that highlights all the modifications we made.

Major changes of the manuscript include a complete restructuring of the content, a new abstract, a shortened introduction part, updated figures, major changes to the discussion section, and a new conclusion section. All these changes were made as already described in our replies to the reviewers' comments except for a few instances where the respective paragraphs were omitted. Please refer to our previous replies and the tracked-changes document for a complete overview of all manuscript modifications.

We are convinced that our revised manuscript represents a major improvement over the previous version and thank the reviewers and the editor for their helpful feedback.

Kind regards,

the authors

[revised manuscript text omitted]

DTM                          Grid data                          LiDAR-based    bare    earth                          AT–Federal state of Lower
                                                                model with 10m resolution                          Austria

---

## Author Response (AR2)

Dear reviewers, dear editor,

Many thanks for your comments and suggestions. Based on your remarks, we have prepared a revised version of our manuscripts in which all issues raised by you have been resolved. Below, you find a point by point overview on our changes.

In addition, we provide a tracked changes document highlighting all manuscript modifications.

Apart from the requested changes, we have added a second affiliation of our third author Benni Thiebes (German Committee for Disaster Reduction – DKKV).

Thanks again for your valuable comments and remarks that helped to improve our manuscript.

Best regards,

The Authors

**Editor comments:**

- All highlighted typos have been corrected.
- We updated the NHESSD publications with their NHESS counterparts where possible.
- We performed a spell check.
- P12 L8: Operative national scale landslide early warning systems
    - Yes, this has also been mentioned by another reviewer and the text has been changed to reflect this. In addition, we added an appropriate reference for an overview on current operational systems.
    On a sidenote, the last I have checked, warnings by the Sri Lanka system used to consist of a handwritten note send by FAX and then scanned as PDF and uploaded onto the website.
- P3 L15: Please define NWP.
    - The abbreviation NWP for Numerical Weather Prediction is already introduced on the previous page.
- P10 L 14: "tricky"
    - We changed the wording.
- P9 L19: less specific
    - We rephrased the sentence.
- P9 L21: circular sentence
    - We rephrased the sentence.
- P10 L18: property zones
    - Yes, you are right. We deleted the term as we think it is not necessary to mention this – it is a common feature of physically-based simulation models
- P13 L6: Changing chapter order
    - The chapter numbering has changed between manuscript iterations. We agree with your suggestion and changed chapter 3 to 2.1. In addition, the header for chapter 2 was modified to better reflect its content, and the numbering of the subsequent sections was changed.
- P19 L12: HPC computing and landslides
    - Many thanks for this hint – we have included the reference.
- P32: X in resolution

- o We have changed this to "Historic 3 hour rainfall event based on spatial interpolation of measurements"

**Reviewer 1:**

- P2 L19: recent review of rainfall thresholds
    - o The suggested reference has been added and fits well, thank you.
- P4 L19 and P2 L9: national scale landslide early warning system
    - o The suggested reference has been added.
    - o The sentence this comment referred to was modified.

**Reviewer 2**

- Proofreading and typographical errors
    - o We have again checked for mistakes and used a spell checker.

[revised manuscript text omitted]
179, 228–247, doi:10.1016/j.earscirev.2018.02.013, 2018.

Piciullo, L., Gariano, S. L., Melillo, M., Brunetti, M. T., Peruccacci, S., Guzzetti, F. and Calvello, M.: Definition and performance of a threshold-based regional early warning model for rainfall-induced landslides, Landslides, 14(3), 995–1008, doi:10.1007/s10346-016-0750-2, 2017.

Pradhan, A. M. S., Kang, H.-S., Lee, J.-S. and Kim, Y.-T.: An ensemble landslide hazard model incorporating rainfall threshold for Mt. Umyeon, South Korea, Bulletin of Engineering Geology and the Environment, doi:10.1007/s10064-017-1055-y, 2017.

R Core Team: R: A Language and Environment for Statistical Computing, R Foundation for Statistical Computing, Vienna, Austria. Available at: https://www.R-project.org, last access: 30 November 2017.

Raia, S., Alvioli, M., Rossi, M., Baum, R. L., Godt, J. W. and Guzzetti, F.: Improving predictive power of physically based rainfall-induced shallow landslide models: a probabilistic approach, Geoscientific Model Development, 7(2), 495–514, doi:10.5194/gmd-7-495-2014, 2014.

Reichenbach, P., Cardinali, M., De Vita, P. and Guzzetti, F.: Regional hydrological thresholds for landslides and floods in the Tiber River Basin (central Italy), Environmental Geology, 35(2–3), 146–159, doi:10.1007/s002540050301, 1998.

Reichle, R. H.: Data assimilation methods in the Earth sciences, Advances in Water Resources, 31(11), 1411–1418, doi:10.1016/j.advwatres.2008.01.001, 2008.

Richwien, W., Lesny, K., 2004. Bodenmechanisches Praktikum, Auswahl und Anwendung von bodenmechanischen Laborversuchen. 11 ed., Verlag Gluckauf GmbH, Essen.

Rossi, G., Catani, F., Leoni, L., Segoni, S. and Tofani, V.: HIRESSS: a physically based slope stability simulator for HPC applications, Nat. Hazards Earth Syst. Sci., 13(1), 151–166, doi:10.5194/nhess-13-151-2013, 2013.

Rossi, M., Peruccacci, S., Brunetti, M. T., Marchesini, I., Luciani, S., Ardizzone, F., Balducci, V., Bianchi, C., Cardinali, M., Fiorucci, F., Mondini, A. C., Reichenbach, P., Salvati, P., Santangelo, M. A., Bartolini, D., Gariano, S. L., Palladino, M., Vessia, G., Viero, A., Antronico, L., Borselli, L., Deganutti, A. M., Iovine, G., Luino, F., Parise, M., Polemio, M., Guzzetti, F., Luciani, S., Fiorucci, F., Mondini, Santangelo, N. and Tonellid, G.: SANF: National warning system for rainfall-induced landslides in Italy, in Proceedings of the 11th International & 2nd North American Symposium on Landslides, vol. 2, edited by E. Eberhardt, C. R. Froese, A. K. Turner, and S. Leroueil, pp. 1895–1899, Taylor & Francis, London., 2012.

Rubio, E., Hall, J. W. and Anderson, M. G.: Uncertainty analysis in a slope hydrology and stability model using probabilistic and imprecise information, Comput. Geotech., 31(7), 529–536, doi:10.1016/j.compgeo.2004.09.002, 2004.

Salciarini, D., Fanelli, G. and Tamagnini, C.: A probabilistic model for rainfall–induced shallow landslide prediction at the regional scale, Landslides, doi:10.1007/s10346-017-0812-0, 2017.

Schaake, J. C., Hamill, T. M., Buizza, R. and Clark, M.: HEPEX: The Hydrological Ensemble Prediction Experiment, Bulletin of the American Meteorological Society, 88(10), 1541–1547, doi:10.1175/BAMS-88-10-1541, 2007.

Schmidt, J., Turek, G., Clark, M. P., Uddstrom, M. and Dymond, J. R.: Probabilistic forecasting of shallow, rainfall-triggered landslides using real-time numerical weather predictions, Natural Hazards and Earth System Science, 8(2), 349–357, 2008.

Schweigl, J. and Hervás, J.: Landslide mapping in Austria, European Commission Joint Research Centre, Institute for Environment and Sustainability, Italy. Available at: https://esdac.jrc.ec.europa.eu/ESDB_Archive/eusoils_docs/Images/EUR23785EN.pdf, last access: 30 November 2017, 2009.

Schwenk, H.: Massenbewegungen in Niederösterreich 1953-1990, Jahrbuch der Geologischen Bundesanstalt, 135(2), 597–660, 1992.

Segoni, S., Piciullo, L. and Gariano, S. L.: A review of the recent literature on rainfall thresholds for landslide occurrence, Landslides, 15(8), 1483–1501, doi:10.1007/s10346-018-0966-4, 2018a.

[revised manuscript text omitted]